# Provably Reliable Conformal Prediction Sets in the Presence of Data Poisoning

**Yan Scholten, Stephan Günnemann**
Department of Computer Science & Munich Data Science Institute
Technical University of Munich
{y.scholten, s.guennemann}@tum.de

## Abstract

Conformal prediction provides model-agnostic and distribution-free uncertainty quantification through prediction sets that are guaranteed to include the ground truth with any user-specified probability. Yet, conformal prediction is not reliable under poisoning attacks where adversaries manipulate both training and calibration data, which can significantly alter prediction sets in practice. As a solution, we propose *reliable prediction sets* (RPS): the first efficient method for constructing conformal prediction sets with provable reliability guarantees under poisoning. To ensure reliability under training poisoning, we introduce smoothed score functions that reliably aggregate predictions of classifiers trained on distinct partitions of the training data. To ensure reliability under calibration poisoning, we construct multiple prediction sets, each calibrated on distinct subsets of the calibration data. We then aggregate them into a majority prediction set, which includes a class only if it appears in a majority of the individual sets. Both proposed aggregations mitigate the influence of datapoints in the training and calibration data on the final prediction set. We experimentally validate our approach on image classification tasks, achieving strong reliability while maintaining utility and preserving coverage on clean data. Overall, our approach represents an important step towards more trustworthy uncertainty quantification in the presence of data poisoning.[1]

## 1 Introduction

Conformal prediction has emerged as a powerful, model-agnostic framework for distribution-free uncertainty quantification. By constructing prediction sets calibrating on hold-out data, it transforms any black-box classifier into a predictor with formal coverage guarantees, ensuring its prediction sets cover the ground truth with any user-specified probability (Angelopoulos & Bates, 2021). This makes it highly relevant for safety-critical applications like medical diagnosis (Vazquez & Facelli, 2022), autonomous driving (Lindemann et al., 2023), and flood forecasting (Auer et al., 2023).

However in practice, noise, incomplete data or adversarial perturbations can lead to unreliable prediction sets (Liu et al., 2024). In particular data poisoning – where adversaries modify the training or calibration data (e.g. during data labeling) – can significantly alter the prediction sets, resulting in overly conservative or empty sets (Li et al., 2024). This vulnerability can undermine the practical utility of conformal prediction in safety-critical applications, raising the research question:

*How can we make conformal prediction sets provably reliable in the presence of data poisoning?*

As a solution, we propose *reliable prediction sets* (RPS): the first efficient method for constructing prediction sets more reliable under data poisoning where adversaries can modify, add and delete datapoints from training and calibration sets. Our approach consists of two key components (Figure 1): First (**i**), we introduce smoothed score functions that reliably aggregate predictions from classifiers trained on distinct partitions of the training data, improving reliability under training poisoning. Second (**ii**), we calibrate multiple prediction sets on disjoint subsets of the calibration data and construct a majority prediction set that includes classes only when a majority of the independent prediction sets agree, improving reliability under calibration poisoning. Using both strategies (**i**) and (**ii**), RPS effectively reduces the influence of individual datapoints during training and calibration.

---

[1] Project page: https://www.cs.cit.tum.de/daml/reliable-conformal-prediction/

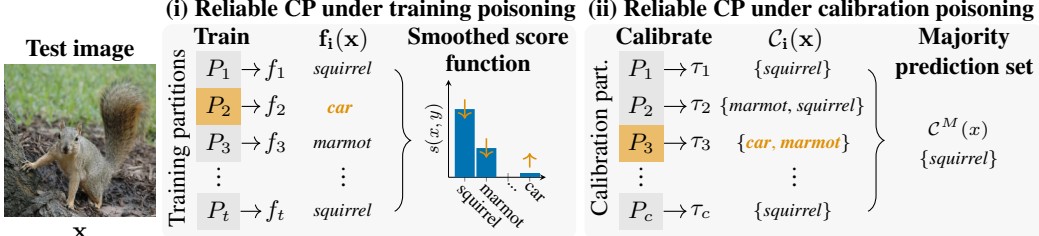

Figure 1: Conformal prediction (CP) is not reliable under poisoning (orange) of training and calibration data, undermining its practical utility in safety-critical applications. As a solution, we propose *reliable prediction sets* (RPS): A novel approach for constructing more reliable prediction sets. We **(i)** aggregate predictions of classifiers trained on distinct partitions, and **(ii)** merge multiple prediction sets $\mathcal{C}_i(x) = \{y : s(x, y) \geq \tau_i\}$ calibrated on separate partitions into a majority prediction set that includes classes only if a majority of the prediction sets $\mathcal{C}_i$ agree. This way RPS reduces the influence of datapoints while preserving the coverage guarantee of conformal prediction on clean data.

We further derive certificates, i.e. provable guarantees for the reliability of RPS under worst-case poisoning. We experimentally validate the effectiveness of our approach on image classification tasks, demonstrating strong reliability under worst-case poisoning while maintaining utility and empirically preserving the coverage guarantee of prediction sets on clean data. Our main contributions are:

- We propose *reliable prediction sets* (RPS) – the first scalable and efficient method for making conformal prediction more reliable under training and calibration poisoning.
- We derive novel certificates that guarantee the reliability of RPS under worst-case data poisoning attacks, including guarantees against label flipping attacks.
- We thoroughly evaluate our method and verify our theorems on image classification tasks.

## 2 RELATED WORK

**Prediction set ensembles.** Ensembles of prediction sets are studied in the uncertainty set literature beyond machine learning (Cherubin, 2019; Solari & Djordjilović, 2022; Gasparin & Ramdas, 2024), e.g. to reduce the effect of randomness. Our work instead proposes a method to improve the reliability of conformal prediction under worst-case training and calibration poisoning.

**Conformal prediction under evasion.** Most works regarding reliable conformal prediction focus on evasion threat models, i.e. adversarial perturbations of the test data. They typically build upon randomized smoothing (Cohen et al., 2019) to certify robustness against evasion attacks (Gendler et al., 2022; Yan et al., 2024; Zargarbashi et al., 2024), or use neural network-specific verification (Jeary et al., 2024). Ghosh et al. (2023) introduce a probabilistic notion as an alternative to worst-case evasion attacks. Unlike prior work on evasion, we consider poisoning threat models.

**Conformal prediction under poisoning.** Despite emerging poisoning attacks (Li et al., 2024), the few existing attempts to improve reliability consider other reliability notions. Most works only certify the coverage guarantee under calibration poisoning (Park et al., 2023; Zargarbashi et al., 2024; Kang et al., 2024). Others study calibration poisoning empirically (Einbinder et al., 2022), under specific label noise (Penso & Goldberger, 2024), or consider distribution shifts between calibration and test data (Cauchois et al., 2020). Zargarbashi et al. (2024) consider modifications to the calibration data, but their threat model does not support adversarial data insertion or deletion. Overall, none of the existing approaches considers pointwise reliability of prediction sets under threat models where adversaries can modify, add or remove datapoints from both training and calibration data.

**Robustness certification against data poisoning.** Most certification techniques for robust classification under poisoning consider other threat models, specific training techniques or architectures (Rosenfeld et al., 2020; Tian et al., 2023; Sosnin et al., 2024). The strongest guarantees also partition the training data and aggregate predictions of classifiers trained on each partition (Levine & Feizi, 2021; Wang et al., 2022; Rezaei et al., 2023). However, all of the prior works only guarantee robust classification and are not directly applicable to certify conformal prediction since prediction sets (1) contain multiple classes, and (2) can be manipulated via poisoning during training and calibration.

## 3 BACKGROUND AND PRELIMINARIES

We focus on classification tasks defined on an input space $\mathcal{X} = \mathbb{R}^d$ for a given finite set of classes $\mathcal{Y} = \{1, \ldots, K\}$. We model prediction set predictors as functions $\mathcal{C} : \mathcal{X} \to 2^{\mathcal{Y}}$, which provide prediction sets as subsets $\mathcal{C}(x) \subseteq \mathcal{Y}$ of the classes $\mathcal{Y}$ for any given datapoint $x \in \mathcal{X}$.

**Exchangeability.** Conformal prediction is a model-agnostic and distribution-free method for constructing prediction sets. It only requires that calibration and test points are exchangeable, which means that their joint distribution is invariant under permutations. In this paper, we adopt the standard assumption (e.g. in image classification) that datapoints are i.i.d., which implies exchangeability. Specifically, we assume three datasets with datapoints sampled i.i.d. from the same distribution $\mathcal{D}$ over $\mathcal{X} \times \mathcal{Y}$: training set $\mathcal{D}_{train}$, calibration set $\mathcal{D}_{calib} = \{(x_i, y_i)\}_{i=1}^n$ and test set $\mathcal{D}_{test}$.

**Conformal prediction.** Conformal prediction transforms any given black-box classifier $f : \mathcal{X} \to \mathcal{Y}$ into a prediction set predictor. We focus on split conformal prediction (Papadopoulos et al., 2002; Lei et al., 2018), the most widely used variant in machine learning. First, one trains a classifier $f$ on the training set and defines a score function $s(x, y)$ that measures conformity between samples $x$ and classes $y$. For example, homogeneous prediction sets (HPS) use the class probabilities of a given soft classifier, $s(x, y) = f_y(x)$ (Sadinle et al., 2019). Then, one computes conformal scores $S = \{s(x_i, y_i)\}_{i=1}^n$ for samples of the calibration set $\mathcal{D}_{calib}$ using the score function $s$. Finally, one can construct prediction sets with the following coverage guarantee (Vovk et al., 1999; 2005):

**Theorem 1.** *Given user-specified coverage probability $1 - \alpha \in (0, 1)$, test sample $(x_{n+1}, y_{n+1}) \in \mathcal{D}_{test}$ exchangeable with $\mathcal{D}_{calib}$, and a score function $s$, we can construct the following prediction set*

$$\mathcal{C}(x_{n+1}) = \{y \in \mathcal{Y} : s(x_{n+1}, y) \geq \tau\},$$

*which fulfills the following marginal coverage guarantee*

$$\Pr[y_{n+1} \in \mathcal{C}(x_{n+1})] \geq 1 - \alpha$$

*for $\tau = Quant(\alpha_n; S)$. Specifically, the threshold $\tau$ is chosen as the $\alpha_n$-quantile of the conformal scores $S$ for a finite-sample corrected significance level $\alpha_n = \lfloor \alpha(n+1) - 1 \rfloor / n$.*

## 4 DESIDERATA FOR RELIABLE CONFORMAL PREDICTION

First we want to outline the desired properties that reliable conformal prediction should exhibit, setting clear goals for how uncertainty should be captured by prediction sets under data poisoning.

**Data poisoning.** While exchangeability may hold for the data distribution $\mathcal{D}$ in theory, the labeled data $\mathcal{D}_l = (\mathcal{D}_{train}, \mathcal{D}_{calib})$ can be poisoned to alter prediction sets in practice. We formalize this threat model, i.e. the strength of poisoning attacks, as a ball centered around labeled data:

$$B_{r_t, r_c}(\mathcal{D}_l) = \left\{ \tilde{\mathcal{D}}_l \mid \delta(\tilde{\mathcal{D}}_{train}, \mathcal{D}_{train}) \leq r_t, \delta(\tilde{\mathcal{D}}_{calib}, \mathcal{D}_{calib}) \leq r_c \right\} \tag{1}$$

where $\delta$ is a distance metric between datasets, and $r_t, r_c$ are the radii for training and calibration sets, respectively. Specifically, we define $\delta$ as the number of inserted or deleted datapoints and label flips, modeling feature modification as two perturbations (deletion and insertion): $\delta(\mathcal{D}_1, \mathcal{D}_2) = |\mathcal{D}_1 \ominus \mathcal{D}_2| - |\mathcal{F}(\mathcal{D}_1, \mathcal{D}_2)|$ where $A \ominus B = (A \setminus B) \cup (B \setminus A)$ is the symmetric set difference between two sets $A$ and $B$, $|S|$ denotes the cardinality of a set $S$, and $\mathcal{F}(\mathcal{D}_1, \mathcal{D}_2)$ represents the set of datapoints with label flips $\mathcal{F}(\mathcal{D}_1, \mathcal{D}_2) = \{x \mid \exists y_1 : (x, y_1) \in \mathcal{D}_1 \setminus \mathcal{D}_2, \exists y_2 : (x, y_2) \in \mathcal{D}_2 \setminus \mathcal{D}_1\}$. Note that we count label flips only once, and feature perturbations can be of arbitrary magnitude.

**Reliability under data poisoning.** Given a datapoint $x \in \mathcal{X}$ and a prediction set $\mathcal{C}(x) \subseteq \mathcal{Y}$, we define reliability of conformal prediction sets under data poisoning as follows:

**Definition 1** (Reliability). *We assume that reliability is compromised if adversaries can remove or add a single class from or to the prediction set $\mathcal{C}(x)$ under our threat model (Equation 1). Specifically, we call prediction sets **coverage reliable** if adversaries cannot shrink prediction sets $\mathcal{C}(x)$ by removing classes, and **size reliable** if adversaries cannot inflate prediction sets $\mathcal{C}(x)$ by adding classes. We further call coverage and size reliable prediction sets **robust**.*

Note that while the coverage guarantee (Theorem 1) provides a marginal guarantee over the entire distribution, our notion of coverage reliability is *pointwise*, i.e. applies to each prediction set $\mathcal{C}(x)$.

Accordingly, we propose the following novel desiderata for reliable conformal prediction:

> **Desiderata for reliable conformal prediction under data poisoning**
>
> **I:** Reliable conformal prediction must provide marginal coverage (Theorem 1) for clean data.
> **II:** Reliable prediction sets must be small (comparable to sets without reliability guarantees).
> **III:** Reliable conformal prediction must *provably* ensure reliability of prediction sets (Definition 1) under data addition, deletion, modification and label flipping (Equation 1) up to a radius that meets the application's needs and safety requirements.
> **IV:** Algorithms for constructing reliable prediction sets must be flexible enough to allow for increased reliability given larger training and calibration sets.
> **V:** Algorithms for constructing reliable prediction sets and their guarantees must be computationally efficient, scalable, and reproducible.

While desideratum **I** requires marginal coverage (Theorem 1), desideratum **II** ensures small sets, and together they prevent that reliability can be achieved trivially by predicting empty or full sets. desideratum **III** ensures that reliability must be certifiable, i.e. a provable guarantee under worst-case poisoning. desideratum **IV** requires that algorithms can increase reliability as more data becomes available since practical risks often increase with more data. Finally, desideratum **V** ensures efficiency in deployment, where reproducibility requires stability under recomputation.

## 5 RELIABLE CONFORMAL PREDICTION SETS

Guided by our desiderata for reliable conformal prediction we introduce **reliable conformal prediction sets** (RPS): the first method for provably reliable conformal prediction under training and calibration poisoning (Figure 1). The first component of RPS (**i**) reliably aggregates classifiers trained on $k_t$ disjoint partitions of the training data. The second component of RPS (**ii**) constructs reliable prediction sets by merging sets calibrated separately on $k_c$ disjoint partitions of the calibration data. Intuitively, larger $k_t$ increases reliability against training poisoning and larger $k_c$ increases reliability against calibration poisoning. We provide detailed instructions in Algorithm 1 and Algorithm 2.

### 5.1 CONFORMAL SCORE FUNCTIONS RELIABLE UNDER TRAINING DATA POISONING

First, our goal is to derive a conformal score function that is reliable under poisoning of training data. This is challenging since the score function also has to quantify agreement between samples and classes, and maintain exchangeability of conformal scores between calibration and test data. To overcome this challenge we propose to (1) partition the training data into $k_t$ disjoint sets, (2) train separate classifiers on each partition, and (3) design a score function that counts the number of classifiers voting for a class $y$ given sample $x$. Since deleting or inserting one datapoint from or into the training set only affects a single partition and thus a single classifier, this procedure effectively reduces the influence of datapoints on the score function.

**Training data partitioning.** To prevent that simple reordering of the datasets affects all partitions simultaneously, we have to partition the training data in a way that is invariant to its order. To achieve this we assign datapoints to partitions by using a hash function directly defined on $x$. For example for images, we use the sum of their pixel values. This technique that has been previously shown to induce certifiable robustness in the context of image classification (Levine & Feizi, 2021). Given a deterministic hash function $h : \mathcal{X} \to \mathbb{Z}$ we define the $i$-th partition of the training set as

$$P_i^t = \{(x_j, y_j) \in \mathcal{D}_{train} : h(x_j) \equiv i \,(\mathrm{mod}\, k_t)\}.$$

Then we deterministically train $k_t$ classifiers $f^{(i)} : \mathcal{X} \to \mathcal{Y}$ on all partitions $P_1^t, \ldots, P_{k_t}^t$ separately.

**Smoothed score function.** Now we define our score function that measures agreement between a sample $x$ and class $y$ by counting the number of classifiers $f^{(i)}$ voting for class $y$ given $x$:

$$s(x, y) = \frac{e^{\pi_y(x)}}{\sum_{i=1}^{K} e^{\pi_i(x)}} \quad \text{with} \quad \pi_y(x) = \frac{1}{k_t} \sum_{i=1}^{k_t} \mathbb{1}\{f^{(i)}(x) = y\} \tag{2}$$

where $\pi_y(x)$ is the percentage of classifiers voting for class $y$ given sample $x$, and $K$ is the number of classes. Note that we introduce the additional softmax over class distribution $\pi(x)$ to prevent overly large prediction sets in practice (desideratum **II**, see Section 7).

| **Algorithm 1** Reliable conformal score function | **Algorithm 2** Reliable conformal prediction sets |
|---|---|
| **Input:** $\mathcal{D}_{train}$, $k_t$, deterministic training algo. $T$ | **Input:** $\mathcal{D}_{calib}$, $k_c$, $s$, $\alpha$, $x_{n+1}$ |
| 1: Split $\mathcal{D}_{train}$ into $k_t$ disjoint partitions $P_i^t$ 
     $P_i^t = \{(x_j, y_j) \in \mathcal{D}_{train} : h(x_j) \equiv i \,(\mathrm{mod}\, k_t)\}$ | 1: Split $\mathcal{D}_{calib}$ into $k_c$ disjoint partitions $P_i^c$ 
     $P_i^c = \{(x_j, y_j) \in \mathcal{D}_{calib} : h(x_j) \equiv i \,(\mathrm{mod}\, k_c)\}$ |
| 2: **for** $i = 1$ **to** $k_t$ **do** | 2: **for** $i = 1$ **to** $k_c$ **do** |
| 3:    Train classifier $f^{(i)} = T(P_i^t)$ on partition $P_i^t$ | 3:    Compute scores $S_i = \{s(x_j, y_j)\}_{(x_j, y_j) \in P_i^c}$ |
| 4: Construct the voting function 
     $\pi_y(x) = \frac{1}{k_t} \sum_{i=1}^{k_t} \mathbb{1}\{f^{(i)}(x) = y\}$ | 4:    Compute $\alpha_{n_i}$-quantile $\tau_i$ of scores $S_i$ 
 5:    Construct prediction set for quantile $\tau_i$ 
     $\mathcal{C}_i(x_{n+1}) = \{y : s(x_{n+1}, y) \geq \tau_i\}$ |
| 5: Smooth the voting function 
     $s(x, y) = e^{\pi_y(x)} / (\sum_{i=1}^{K} e^{\pi_i(x)})$ | 6: Construct majority vote prediction set 
     $\mathcal{C}^M(x_{n+1}) = \{y : \sum_{i=1}^{k_c} \mathbb{1}\{y \in \mathcal{C}_i(x_{n+1})\} > \hat{\tau}(\alpha)\}$ |
| **Output:** Reliable conformal score function $s$ | **Output:** Reliable conformal prediction set $\mathcal{C}^M$ |

For any function to be considered a valid score function for conformal prediction it has to maintain exchangeability of conformal scores between calibration and test data (Angelopoulos et al., 2021).

**Lemma 1.** *The smoothed score function in Equation 2 is a valid conformal score function.*

*Proof.* We use one function to score all points independent of other datapoints and which dataset they belong to (and where in the dataset). Thus, given exchangeable data, scores computed by our smoothed score function remain exchangeable. Therefore $s$ of Equation 2 is a valid score function. $\square$

Lemma 1 implies that the coverage guarantee (Theorem 1) holds on clean data when using our smoothed score function (desideratum **I**). Intuitively, our score function quantifies uncertainty by the number of votes from multiple classifiers (instead of the logits of one classifier). As long as classifiers are trained on isolated partitions we can reduce the influence of datapoints on the conformal scores. We summarize instructions for the smoothed score function in Algorithm 1.

## 5.2 MAJORITY PREDICTION SETS RELIABLE UNDER CALIBRATION DATA POISONING

Now we derive prediction sets reliable against calibration poisoning. This is challenging since the prediction sets must also achieve marginal coverage on clean data (desideratum **I**) without inflating set size (desideratum **II**). We propose to (1) partition the calibration data into $k_c$ disjoint sets, (2) compute separate prediction sets based on the conformal scores on each partition, and (3) merge the resulting prediction sets via majority voting. This improves reliability since adversaries have to poison multiple partitions to alter the majority vote. We further show that such majority prediction sets achieve marginal coverage, and do not grow too much in size in practice (Section 7).

**Calibration data partitioning.** We partition the calibration data as follows: Given hash function $h$ we define the $i$-th partition of the calibration set as $P_i^c = \{(x_j, y_j) \in \mathcal{D}_{calib} : h(x_j) \equiv i \,(\mathrm{mod}\, k_c)\}$. We then use a (potentially reliable) conformal score function $s$ to compute the conformal scores $S_i = \{s(x_j, y_j)\}_{(x_j, y_j) \in P_i^c}$ on each partition $P_i^c$. We can then determine the $\alpha_{n_i}$-quantiles of the separate conformal scores, $\tau_i = \mathrm{Quant}(\alpha_{n_i}; S_i)$, where $n_i$ is the size of the $i$-th partition, $n_i = |P_i^c|$.

**Majority prediction sets.** Now we propose prediction sets, which are provably reliable under calibration poisoning. Given a new datapoint $x_{n+1} \in \mathcal{D}_{test}$ we construct $k_c$ prediction sets for each partition as $\mathcal{C}_i(x_{n+1}) = \{y : s(x_{n+1}, y) \geq \tau_i\}$. We then construct a prediction set composed of all classes that appear in the majority of *independent* prediction sets (see instructions in Algorithm 2):

$$\mathcal{C}^M(x_{n+1}) = \left\{ y : \sum_{i=1}^{k_c} \mathbb{1}\{y \in \mathcal{C}_i(x_{n+1})\} > \hat{\tau}(\alpha) \right\} \tag{3}$$

with quantile function $\hat{\tau}(\alpha) = \max\{x \in [k_c] : F(x) \leq \alpha\}$ for $[k_c] = \{0, \ldots, k_c\}$, where $\hat{\tau}(\alpha)$ is the inverse of the CDF $F$ of the Binomial distribution $\mathrm{Bin}(k_c, 1 - \alpha)$. Intuitively, we select the required majority $\hat{\tau}(\alpha)$ such that the sum over $k_c$ Bernoulli random variables $\mathbb{1}\{y \in \mathcal{C}_i(x_{n+1})\}$ (each with success probability at least $1 - \alpha$) is at most $\hat{\tau}(\alpha)$ with probability at most $\alpha$. Note that for $k_c = 1$ we have $\hat{\tau}(\alpha) = 0$, for which $\mathcal{C}^M$ amounts to vanilla conformal prediction. Notably, such majority prediction sets achieve marginal coverage on clean data (Proof in Appendix D):

**Theorem 2.** *Given any conformal score function $s$ and test sample $(x_{n+1}, y_{n+1}) \in \mathcal{D}_{test}$ exchangeable with $\mathcal{D}_{calib}$, the majority prediction set (Equation 3) – constructed from sets calibrated on disjoint partitions – achieves marginal coverage on clean data:* $\Pr[y_{n+1} \in \mathcal{C}^M(x_{n+1})] \geq 1 - \alpha$.

*Proof sketch.* Since each prediction set $\mathcal{C}_i$ fulfills marginal coverage, $\Pr[y_{n+1} \in \mathcal{C}_i(x_{n+1})] \geq 1-\alpha$, the probability of the sum over Bernoulli random variables $\mathbb{1}\{y_{n+1} \in \mathcal{C}_i(x_{n+1})\}$ being larger than $\hat{\tau}(\alpha)$ is at least $1-\alpha$ (due to the selection of $\hat{\tau}(\alpha)$). Thus we have $\Pr[y_{n+1} \in \mathcal{C}^M(x_{n+1})] \geq 1-\alpha$. $\square$

Note that majority voting is also used in the context of merging uncertainty sets (Gasparin & Ramdas, 2024), but their approach comes without reliability guarantees (see discussion in Appendix D). Importantly, Theorem 2 guarantees marginal coverage for any conformal score function. This holds especially for our smoothed score function (Equation 2). As a consequence, majority prediction sets based on the smoothed score function achieve marginal coverage (desideratum **I**).

## 6 PROVABLE GUARANTEES FOR RELIABLE CONFORMAL PREDICTION SETS

After introducing reliable prediction sets (RPS), we derive certificates for their reliability as defined in Definition 1 and required by desideratum **III**. We consider the threat model $B_{r_t, r_c}(\mathcal{D}_l)$ where adversaries can insert, delete and flip labels for up to $r_t$ training and $r_c$ calibration points (Section 3). In the following we treat training poisoning, then calibration poisoning, and finally poisoning of both.

### 6.1 GUARANTEES FOR SMOOTHED SCORING FUNCTION UNDER TRAINING POISONING

We begin with the reliability of the smoothed scoring function under training poisoning ($r_t > 0$, $r_c = 0$). Let $\mathcal{C}(x_{n+1}) = \{y \in \mathcal{Y} : s(x_{n+1}, y) \geq \tau\}$ be a prediction set for a new test point $x_{n+1}$ derived using conformal prediction (Section 3) under the clean dataset $\mathcal{D}_l$ with smoothed score function $s$. Our goal is to bound the prediction set $\tilde{\mathcal{C}}(x_{n+1})$ derived under any poisoned dataset $\tilde{\mathcal{D}}_l \in B_{r_t, r_c}(\mathcal{D}_l)$. This requires that we bound score function $s$ and quantile $\tau$. We start with the score function:

**Lemma 2.** *We can upper bound the score function for any $\tilde{\mathcal{D}}_l \in B_{r_t, r_c}(\mathcal{D}_l)$ as follows:*

$$\overline{s}(x, y) = \max_{\substack{0 \leq \pi_i \leq 1 \\ \Delta_i \in \{0, \pm\frac{1}{k_t}, \ldots, \pm\frac{r_t}{k_t}\} \\ \sum_{i=1}^{K} \Delta_i = 0}} \frac{e^{\pi_y}}{\sum_{i=1}^{K} e^{\pi_i}} \quad \text{with} \quad \pi = [\pi_1(x) + \Delta_1, \ldots, \pi_K(x) + \Delta_K] \quad (4)$$

Proof in Appendix E. Although optimizing softmax functions typically leads to non-convex optimization problems, the problem in (4) reduces to a discrete optimization problem that can be solved efficiently (desideratum **V**). We derive algorithms computing lower and upper bounds in $r_t$ steps, presenting the upper bound in Algorithm 3. Intuitively, in each step we greedily redistribute $\frac{1}{k_t}$ probability mass from the current class $\hat{y} \neq y$ with the largest probability mass to the target class $y$. We repeat this process until we have redistributed the entire probability mass $\frac{r_t}{k_t}$. We present the lower bound algorithm and proofs in Appendix E.

---

**Algorithm 3** Greedy algorithm for upper bounding the smoothed score function $s$

**Input:** Score function $s$, $x$, $y$, $k_t$, $r_t$
1: $\pi = [\pi_1(x), \ldots, \pi_K(x)]$
2: **for** $i = 1$ **to** $r_t$ **do**
3: $\quad \hat{y} = \text{argmax}_{\hat{y} \neq y} \pi_{\hat{y}}$
4: $\quad \pi_{\hat{y}} \leftarrow \min(\max(\pi_{\hat{y}} - 1/k_t, 0), 1)$
5: $\quad \pi_y \leftarrow \min(\max(\pi_y + 1/k_t, 0), 1)$
**Output:** $\overline{s}(x, y) = e^{\pi_y} / (\sum_{i=1}^{K} e^{\pi_i})$

---

Given lower and upper bounds $\underline{z_i} = \underline{s}(x_i, y_i)$ and $\overline{z_i} = \overline{s}(x_i, y_i)$ on the conformal scores for all points $(x_i, y_i) \in \mathcal{D}_{calib}$ in the calibration set, we can directly determine the worst-case quantiles:

$$\underline{\tau} = \text{Quant}(\alpha_n; \{\underline{z_i}\}_{i=1}^n) \qquad \overline{\tau} = \text{Quant}(\alpha_n; \{\overline{z_i}\}_{i=1}^n)$$

Finally we need to identify (1) the class within the prediction set that received the fewest votes from the classifiers $f^{(i)}$ and (2) the class outside the prediction set that got most votes from the classifiers:

$$\underline{y} = \text{argmin}_{y \in \mathcal{C}(x_{n+1})} \pi_y(x) \qquad \overline{y} = \text{argmax}_{y \notin \mathcal{C}(x_{n+1})} \pi_y(x)$$

Then we can provide the following guarantees (Proof in Appendix E):

**Theorem 3.** *Given $r_c = 0$, the conformal prediction set $\tilde{\mathcal{C}}(x_{n+1})$ derived with the smoothed score function under any poisoned dataset $\tilde{\mathcal{D}}_l \in B_{r_t, r_c}(\mathcal{D}_l)$ is coverage reliable, i.e. $\tilde{\mathcal{C}}(x_{n+1}) \supseteq \mathcal{C}(x_{n+1})$, if $\underline{s}(x, \underline{y}) \geq \overline{\tau}$, and size reliable, i.e. $\tilde{\mathcal{C}}(x_{n+1}) \subseteq \mathcal{C}(x_{n+1})$, if $\overline{s}(x, \overline{y}) < \underline{\tau}$.*

Intuitively, if class $\underline{y}$ cannot be removed from $\mathcal{C}(x_{n+1})$ ($\overline{y}$ added), adversaries cannot remove (add) any other class and thus the prediction sets are coverage (size) reliable.

## 6.2 GUARANTEES FOR MAJORITY PREDICTION SETS UNDER CALIBRATION POISONING

Now we analyze reliability of our majority prediction sets under calibration poisoning ($r_t=0, r_c>0$). Let $\mathcal{C}^M(x_{n+1})$ be the majority prediction set for a new test point $x_{n+1}$ derived under the clean dataset $\mathcal{D}_l$ using any deterministic conformal score function $s$. Intuitively, if adversaries cannot remove (add) a class from the majority prediction set by removing (adding) it from (to) $r_c$ individual prediction sets, the majority prediction set remains coverage (size) reliable even in the worst case. This is since adversaries can perturb at most $r_c$ calibration partitions under our threat model. To determine whether adversaries can remove or add classes, we have to count minimum and maximum support $\sum_{i=1}^{k_c} \mathbb{1}\{y \in \mathcal{C}_i(x_{n+1})\}$ for classes in and outside of the majority set:

$$\underline{m} = \min_{y \in \mathcal{C}^M} \sum_{i=1}^{k_c} \mathbb{1}\{y \in \mathcal{C}_i(x_{n+1})\} \qquad \overline{m} = \max_{y \notin \mathcal{C}^M} \sum_{i=1}^{k_c} \mathbb{1}\{y \in \mathcal{C}_i(x_{n+1})\}$$

Using each support we can provide the following guarantees (Proof in Appendix E):

**Theorem 4.** *Given $r_t=0$ and deterministic score function $s$, the majority prediction set $\tilde{\mathcal{C}}^M(x_{n+1})$ derived under any dataset $\tilde{\mathcal{D}}_l \in B_{r_t,r_c}(\mathcal{D}_l)$ is coverage reliable, i.e. $\tilde{\mathcal{C}}^M(x_{n+1}) \supseteq \mathcal{C}^M(x_{n+1})$, if $\underline{m} - r_c > \hat{\tau}(\alpha)$, and size reliable, i.e. $\tilde{\mathcal{C}}^M(x_{n+1}) \subseteq \mathcal{C}^M(x_{n+1})$, if $\overline{m} + r_c \leq \hat{\tau}(\alpha)$, provided that the smallest calibration partition $i^*$ is large enough $|P_{i^*}^c| - r_c \geq (\frac{1}{\alpha} - 1)$.*

Note that the last condition ensures that the calibration partitions are large enough such that worst-case adversaries cannot delete datapoints to prevent us from computing the majority prediction sets.

## 6.3 PROVABLE RELIABILITY GUARANTEES FOR RPS UNDER GENERAL DATA POISONING

Finally, we consider poisoning of training and calibration data ($r_t > 0, r_c > 0$).

**Coverage reliability.** To ensure coverage reliability we have to show that all classes $y \in \mathcal{C}^M$ are guaranteed to be in the majority prediction set under worst-case poisoning. The majority prediction set $\mathcal{C}^M$ contains a class $y$ only if it appears in a majority of $\hat{\tau}(\alpha)$ individual prediction sets $\mathcal{C}_i$. Under calibration poisoning, adversaries can remove classes from $r_c$ individual prediction sets. Intuitively, the number of prediction sets reliable under training poisoning $\beta_y$ must be large enough such that even under calibration poisoning, the number of prediction sets containing the class is still larger than the threshold, $\beta_y - r_c > \hat{\tau}(\alpha)$. This leads to the following guarantee (Proof in Appendix E):

**Theorem 5.** *Let $\beta_y$ denote the number of prediction sets $\mathcal{C}_i \in \{\mathcal{C}_1, \ldots, \mathcal{C}_{k_c}\}$ for which we can guarantee $y \in \mathcal{C}_i$ under $r_t$ poisoned training datapoints. If $\beta_y - r_c > \hat{\tau}(\alpha)$ for all $y \in \mathcal{C}^M(x_{n+1})$ then the majority prediction set is coverage reliable under any dataset $\tilde{\mathcal{D}}_l \in B_{r_t,r_c}(\mathcal{D}_l)$, provided that the smallest calibration partition $i^*$ is large enough $|P_{i^*}^c| - r_c \geq (\frac{1}{\alpha} - 1)$.*

**Size reliability.** To ensure size reliability we have to show that all classes $y \notin \mathcal{C}^M$ are guaranteed to stay outside of the majority prediction set under worst-case poisoning. The majority prediction set $\mathcal{C}^M$ does not contain a class $y$ if it appears in less than or equal to $\hat{\tau}(\alpha)$ individual prediction sets $\mathcal{C}_i$. Under calibration poisoning, adversaries can add classes to $r_c$ prediction sets in the worst-case. Intuitively, if we can guarantee that $\gamma_y$ prediction sets $\mathcal{C}_i$ do not contain the class $y$ under training poisoning, at most $k_c - \gamma_y$ prediction sets contain the class under worst-case training poisoning. This number of prediction sets containing the class in the worst-case must be small enough such that even if adversaries add the class to $r_c$ prediction sets, the majority prediction set does not contain the class, $k_c - \gamma_y + r_c \leq \hat{\tau}(\alpha)$. This leads to the following guarantee (Proof in Appendix E):

**Theorem 6.** *Let $\gamma_y$ denote the number of prediction sets $\mathcal{C}_i \in \{\mathcal{C}_1, \ldots, \mathcal{C}_{k_c}\}$ for which we can guarantee $y \notin \mathcal{C}_i$ under $r_t$ poisoned training datapoints. If $k_c - \gamma_y + r_c \leq \hat{\tau}(\alpha)$ for all $y \notin \mathcal{C}^M(x_{n+1})$ then the majority prediction set is size reliable under any dataset $\tilde{\mathcal{D}}_l \in B_{r_t,r_c}(\mathcal{D}_l)$, provided that the smallest calibration partition $i^*$ is large enough $|P_{i^*}^c| - r_c \geq (\frac{1}{\alpha} - 1)$.*

Note that we can efficiently compute numbers $\beta_y$ and $\gamma_y$ as described in Subsection 6.1 by computing the worst-case quantiles and verifying $\underline{s}(x,y) < \overline{\tau}$ and $\overline{s}(x,y) < \underline{\tau}$, respectively. Our overall certification approach is efficient in practice (desideratum **V**) as we discuss in Appendix E and experimentally demonstrate in the next section.

## 7 EXPERIMENTAL EVALUATION

In this section we evaluate our reliable prediction sets and their worst-case guarantees, demonstrating their effectiveness in analyzing and improving reliability of conformal prediction under poisoning. We compare three settings (calibration poisoning, training poisoning, and poisoning of both) by computing the following prediction sets: (a) majority prediction sets merging multiple homogeneous prediction sets calibrated on each partition, (b) conformal prediction sets using our smoothed score function, and (c) majority prediction sets using our smoothed score function.

**Datasets and models.** We train ResNet18, ResNet50 and ResNet101 models (He et al., 2016) on SVHN (Netzer et al., 2011), CIFAR10 and CIFAR100 (Krizhevsky et al., 2009). The datasets contain images with 3 channels of size 32x32, categorized into 10, 10 and 100 classes, respectively. Unless stated otherwise, we show results for ResNet18 on CIFAR10 and coverage level $1 - \alpha = 0.9$ in this section and provide additional experimental results in Appendix B.

**Experimental setup.** We randomly select 1,000 images of the test set for calibration and use the remaining 9,000 datapoints for testing. To account for randomness in model initialization and calibration set sampling, we train 5 classifiers with different initializations and validate each of them on 5 different calibration splits. We report mean and standard deviation (shaded areas in the plots). We refer to Appendix A for the full experimental setup including detailed reproducibility instructions.

**Evaluation metrics.** We report three *reliability ratios*: The ratios of test datapoints whose prediction sets are, according to our worst-case analysis, coverage reliable (classes cannot be removed), size reliable (classes cannot be added), or robust (classes cannot removed or added). *Empirical coverage* refers to the ratio of datapoints whose prediction sets cover the ground truth label of the test set. We also report the *average size* of the prediction sets computed on the test set.

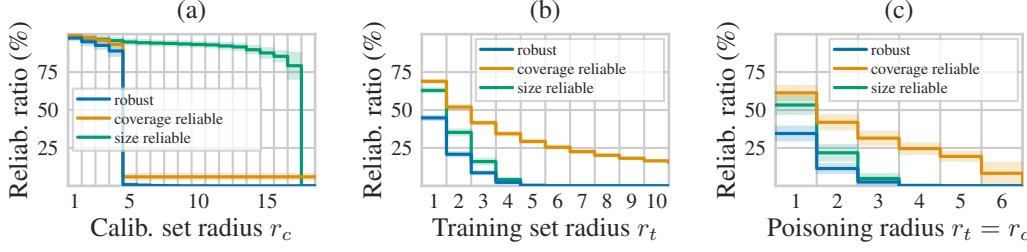

Figure 2: Worst-case reliability guarantees across three scenarios: (a) poisoning of the calibration data, (b) poisoning of the training data, and (c) poisoning of both datasets. Our guarantees against coverage attacks are stronger when training data is poisoned, whereas for calibration attacks our method offers stronger guarantees for size reliability. Notably, even under strong adversarial conditions where both datasets can be poisoned we still provide non-trivial reliability guarantees.

**(a) Reliability of majority prediction sets under calibration poisoning.** Majority prediction sets demonstrate strong reliability guarantees against calibration set poisoning in empirical evaluations (Figure 2 a). Specifically, we construct majority prediction sets by merging $k_c$=22 homogeneous prediction sets, each calibrated on separate calibration partitions, resulting in an empirical coverage of 90.2% and an average set size of 0.94. When up to $r_c$=4 datapoints in the calibration set are poisoned, our method guarantees that over 93% of the prediction sets remain reliable against worst-case coverage attacks. The guarantees against set size attacks are even stronger: Even if $r_c$=15 datapoints are poisoned we still guarantee that over 87% of the prediction sets remain size reliable.

**(b) Reliability of smoothed score function under training poisoning.** The setting of training set poisoning is considerably more challenging since adversaries can simultaneously manipulate the quantiles during calibration and the scores at inference. We compute conformal prediction sets using our smoothed score function on $k_t$=100 training partitions, resulting in empirical coverage of 90.7% and average set size of 3.18 (Figure 2 b). Despite strong adversaries, our reliable prediction sets still manage to provide non-trivial reliability guarantees under worst-case perturbations. Specifically, when up to $r_t = 4$ datapoints in the training set are poisoned, our method guarantees that over 34% of the prediction sets remain reliable against worst-case coverage attacks.

**(c) Reliability of RPS under training and calibration poisoning.** By far the most challenging setting constitutes adversaries that manipulate both training and calibration data. We compute majority prediction sets using our smoothed score function on $k_t$=100 training partitions and $k_c$=40 calibration partitions, resulting in empirical coverage of 92% and average set size of 3.41 (Figure 2 c). Notably, under poisoning of up to $r_t = 3$ training *and* $r_c = 3$ calibration points, our method still guarantees that over 31% of the prediction sets remain reliable against worst-case coverage attacks.

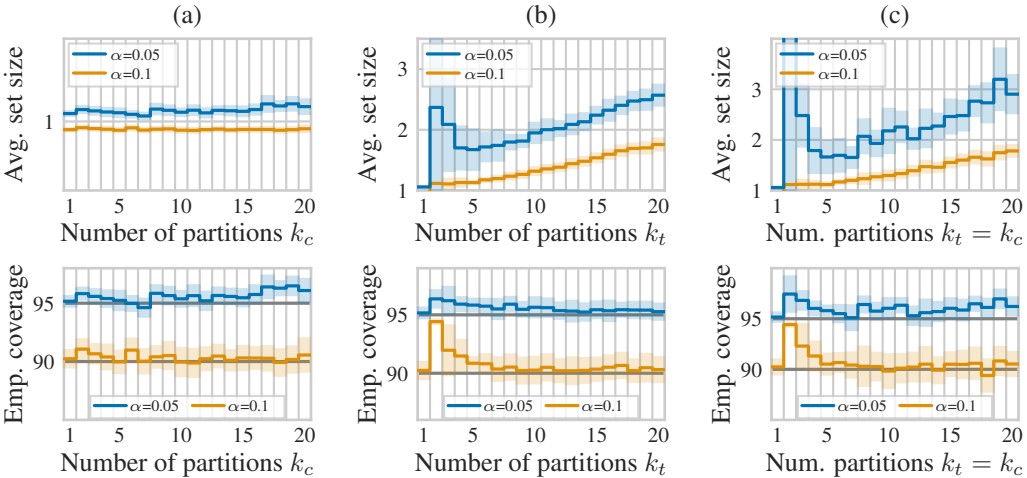

Figure 3: Average set size and empirical coverage in all three different experiment settings (a–c). Notably, our reliable prediction sets yield valid coverage guarantees without becoming too large.

**Average set size.** In Figure 3 (top row) we study the average set size. Notably, our majority prediction sets yield strong guarantees (Figure 2 a) without any significant increase in size (Figure 3 a): the average prediction set size remains below 1, which even holds for $k_c$=40 ($\alpha$=0.1). Interestingly, the size increases with more training partitions, creating a tade-off between reliability and utility of the prediction sets. Practitioners have to fine-tune this trade-off depending on the application's sensitivity. Additionally, we study set sizes on different datasets under calibration poisoning (setting (a)): On CIFAR100 the average set size remains below 4 for $k_c \leq 26$, demonstrating that our majority prediction sets scale well to datasets with more classes (Figure 4 (1)). Overall, we empirically find that our reliable prediction sets do not become too large in size (desideratum **II**).

**Empirical coverage.** In Figure 3 (bottom row) we empirically validate the coverage guarantee (Theorem 2) on clean data (desideratum **I**). In Subsection B.1 we provide additional empirical evidence that, while concentration around the nominal level $1 - \alpha$ naturally decreases for smaller sets, majority prediction sets are again closely concentrated around the nominal coverage level.

**Minimal number of classifiers.** We observe increasing empirical coverage and sizes when using only $k_t$=2 or 3 classifiers (spikes in Figure 3). Intuitively, a small number of classifiers makes the smoothed score function less stable during calibration. In practice, a sufficient number of classifiers is required to achieve consistent majority vote consensus and reliability. Notably, our analysis shows that four classifiers are already sufficient to prevent excessively large prediction sets.

**Softmax ablation study.** We found that smoothing the voting function with a softmax (Section 5) avoids overly large prediction sets and overcoverage in practice. To demonstrate this we conduct an experiment (Figure 4 (2, 3)) for varying numbers of training partitions $k_t$, where we compare conformal prediction with our smoothed score function $s$ against using the voting function $\pi$ only.

**Computational efficiency.** Training the classifiers takes most of the time (statistics in Appendix A). Note, however, that while having to train more classifiers, each one is trained on a subset of the training data, which can speed up the training process. Inference with the ResNet18 models takes between 2 and 10 seconds on CIFAR10. Constructing the conformal prediction sets takes at most 0.5 seconds for the entire test set. Computing certificates for majority prediction sets takes less than a second, and around one minute when computing guarantees under training and calibration poisoning. Overall, we found that reliable prediction sets are computationally efficient (desideratum **V**).

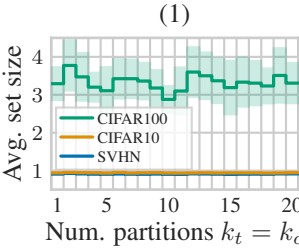 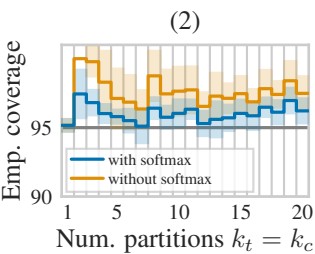 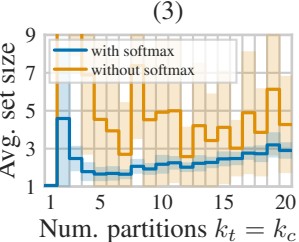

Figure 4: (1): Average prediction set size of majority prediction sets across three different datasets. (2,3): Softmax ablation study for empirically justifying smoothing of the voting function ($\alpha = 0.05$).

## 8 DISCUSSION

**Reproducible prediction sets.** Score functions in the literature are not just unreliable but some also depend on randomization to break ties (Romano et al., 2020). While randomization at inference changes prediction sets by at most one class (Angelopoulos et al., 2021), different prediction sets for the same input may not be desirable from a reliability standpoint, violating desideratum **V**. For example, a differential diagnosis for two patients with identical health should not yield different results. As a remedy, we propose to make randomized score functions reproducible (Appendix C).

**Limitations.** Although RPS computationally scales to larger settings, training on subsets of larger datasets such as CIFAR100 or ImageNet comes with accuracy loss, which also affects the utility of our smoothed score function (details in Appendix B). This accuracy loss is an open challenge in the general certifiably robust classification literature and beyond the scope of this paper.

**Minimal calibration set size for majority prediction sets.** Recall that desideratum **VI** requires that the reliability of prediction sets must increase for larger calibration sets. Our majority prediction sets fulfill this desideratum by construction since increasing the number of partitions $k_c$ will decrease the influence of datapoints. However, we need enough data to construct our prediction sets: due to the finite-sample correction, the calibration partitions cannot become arbitrarily small. If the hashing function would distribute all calibration images into equally-sized partitions of size $n/k_c$, we would need at least $n \geq k_c \left(\frac{1}{\alpha} - 1\right)$ calibration points in total (Proof in Appendix D). Notably, this relationship is linear: given a fixed coverage probability of $1 - \alpha$, increasing the calibration partitions by a factor of $k$ only requires $k$-times larger calibration sets, which is realistic for all commonly used image classification datasets and coverage probabilities used in the literature.

**Training poisoning discussion.** Under the assumption of exchangeability, adversaries cannot compromise the coverage guarantee of Theorem 1 by poisoning training data. This holds because conformal scores are computed using a fixed classifier (post-training). However, in practice, the exchangeability assumption may not always hold: Adversaries could exploit knowledge of the calibration set to manipulate the training process, causing the classifier to perform differently on the (known) calibration set than on unseen test data. Moreover, even if exchangeability holds, adversaries can still affect individual prediction sets: For example, they could manipulate the training process to degrade the utility of the score function, significantly altering the resulting prediction sets. This underscores again the need for *point-wise* coverage and size reliability as we introduce in Definition 1.

## 9 CONCLUSION

We introduce *reliable prediction sets* (RPS), a novel method designed to improve pointwise reliability of conformal prediction sets in the presence of data poisoning and label flipping attacks. By leveraging smoothed score functions and a majority voting mechanism, RPS effectively mitigates the influence of adversarial perturbations during both training and calibration. We provide theoretical guarantees that RPS maintains stability under worst-case data poisoning, and demonstrate the effectiveness of our approach on image classification tasks. Overall, our approach represents an important contribution towards more reliable uncertainty quantification in practice, fostering the trustworthiness in real-world scenarios where data integrity cannot be guaranteed.

REPRODUCIBILITY STATEMENT

We ensure reproducibility by documenting all hyperparameters and experimental setups in App. A. We also provide code along with detailed reproducibility instructions via the following project page: https://www.cs.cit.tum.de/daml/reliable-conformal-prediction/.

ACKNOWLEDGMENTS

The authors want to thank Jan Schuchardt, Lukas Gosch, and Leo Schwinn for valuable feedback on the manuscript. This work has been funded by the DAAD program Konrad Zuse Schools of Excellence in Artificial Intelligence (sponsored by the Federal Ministry of Education and Research). The authors of this work take full responsibility for its content.

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

APPENDIX OVERVIEW

In this appendix we provide additional results, details on our experimental setup and prove theoretical results as outlined in the following:

## A    FULL EXPERIMENTAL SETUP AND REPRODUCIBILITY DETAILS

We provide details on the experimental setup to ensure reproducibility of our results.

**Datasets.** The datasets we use for evaluation are described in Section 7 (SVHN (Netzer et al., 2011), CIFAR10 and CIFAR100 (Krizhevsky et al., 2009)) and are publicly available. We use the torchvision library to load the datasets.[2]  We normalize images before training, and we compute dataset mean and standard deviation on each training partition separately to ensure models are trained on isolated partitions, which is required by our method to improve reliability.

**Training details.** We train all models with stochastic gradient descent (learning rate 0.01, momentum 0.9, weight decay 5e-4) for 400 epochs using early stopping if the training accuracy does not improve for 100 epochs. We further deploy a cosine learning rate scheduler (Loshchilov & Hutter, 2017). We use a batch size of 128 during training and 300 at inference. To ensure that our guarantees against training-poisoning hold we require that the training process is deterministic, which not only involves fixing the random seed for data augmentation but also ensuring that the internal training processes are deterministic (Levine & Feizi, 2021). We also sort the training data in each partition to ensure that training is invariant w.r.t. the order of the training data (Levine & Feizi, 2021).

**Image preprocessing.** We determine dataset-wide mean and standard deviation dynamically on every training set partition separately once before training to ensure that each classifier is trained on an isolated partition. We subsequently normalize all images in one partition with the corresponding values. We also augment the training set with random crops (padding of 4 pixels) and random horizontal flips (but we perform the data augmentation in a deterministic, reproducible way across runs). Note that mean and standard deviation are computed before data augmentation.

**Hardware details.** We train ResNet18 models on a NVIDIA GTX 1080TI GPU, and the ResNet50 and ResNet101 models on a NVIDIA A100 40GB. We perform inference of all models on a NVIDIA GTX 1080TI GPU, and compute certificates on a Xeon E5-2630 v4 CPU.

**Reproducibility.** To ensure reproducibility we use random seeds for all randomized functions, this especially includes the dataset preprocessing, model training and calibration splits. We provide all random seeds and detailed reproducibility instructions along with our code.

**Training time details.** The average runtime statistics for training ResNet18 models on CIFAR-10 and SVHN are as follows. Training a single ResNet18 model on CIFAR-10 takes approx. 2.6 hours, while training 100 models requires a total of 2.4 hours, with each individual model taking approx. 1.5 minutes. For SVHN, a single ResNet18 model takes 1.5 hours to train, and training 100 models requires a total of 1.6 hours, with each model training taking around 58 seconds. For CIFAR100, a single ResNet18 model takes 3 hours to train, and training 100 models requires a total of 2.27 hours, with each model training for around 1.4 minutes.

---

[2]`https://pytorch.org/vision/stable/index.html`

## B ADDITIONAL RESULTS FOR RELIABLE PREDICTION SETS

In this section, we expand on the experimental results by providing further analyses and complementary results. Figure 5 shows the worst-case reliability guarantees for the SVHN dataset under the three different poisoning scenarios (in the same settings as described in the main paper). We again observe that our method provides reliable prediction sets even under worst-case poisoning attacks. Complementary to the main section, Figure 6 shows the average set size and empirical coverage in all three different experiment settings (a–c) on the SVHN dataset.

Figure 7 (1) shows the average set sizes of the three different architectures (ResNet18, ResNet50, ResNet101) on the CIFAR10 dataset when using majority vote prediction sets with the smoothed score function (third experimental setting). Interestingly, using the ResNet18 model yields the best results, which we attribute to the fact that models trained on subsets of the training set require less capacity to learn the data distribution, and the reduced size prevents overfitting. Figure 7 (2,3) show the softmax ablation study for empirically justifying the smoothing of the voting function ($\alpha = 0.1$).

In the main plots in Section 7, we only considered the diagonal $r_t = r_c$ for the reliability ratios. In Figure 8, we show the reliability ratios for coverage reliability, size reliability, and robustness for all combinations of $r_t$ and $r_c$ for the CIFAR10 dataset ($k_t$=100, $k_c$=40, $\alpha$=0.1). Reliability ratios are generally higher for larger $r_c$, which indicates that RPS is more reliable under calibration poisoning.

In Figure 9 we provide additional results for ResNet18 on the CIFAR100 dataset under calibration poisoning. We show the empirical coverage, average set size, and reliability guarantees for the ResNet18 model. We observe that our method provides reliable prediction sets even under calibration poisoning attacks on the CIFAR100 dataset (showing that majority prediction sets generally scale to datasets with significantly more classes).

Figure 10 shows the relationship between reliability of ResNet18 and the number of partitions under calibration poisoning for SVHN, CIFAR10, and CIFAR100. For training poisoning, guarantees become non-trivial for $k_t = 100$ partitions as shown in the main text. Note that size reliability reduces again if the number of calibration partitions becomes too large. The main reason for this is that a minimum number of calibration points is required to construct prediction sets (due to the finite-sample correction, as already discussed in Section 8). As the number of calibration points per partition decreases with more partitions, adversaries could remove points from partitions to prevent us from constructing prediction sets in the first place (see theoretical analysis in Appendix D).

As mentioned in Section 8, training on subsets of larger datasets such as CIFAR100 or ImageNet comes with accuracy loss, which affects the utility of our smoothed score function. Specifically, when splitting the training set of CIFAR100 into 10 partitions, each individual classifier achieves an accuracy of approximately 30% (in contrast to at least 70% when trained on the entire dataset). This causes inconsistencies between the classifiers, which affects the utility of the score function, eventually leading to excessively large prediction sets. Figure 11 (a) shows that the average set size is above 42 for 10 partitions. When using 100 partitions, the average set size is 42.96, indicating low utility. Future learning algorithms could further improve performance under training poisoning for larger datasets with more classes, ultimately boosting robustness and reliability in machine learning.

In additional experiments (Figure 11) we finetune classifiers pretrained on ImageNet (instead of training randomly initialized models from scratch as for all other plots in the paper). Note that in this case the reliability guarantees hold for the fine-tuning dataset only, not for the dataset used for pretraining. Interestingly, Figure 11 (1) shows that the average set size in experiments with pretrained models is significantly reduced, indicating that the models benefit from pretraining on ImageNet (although the average size still remains high for 100 partitions on CIFAR100 – 18.16). Further, Figure 11 (2-3) show reliability guarantees under both training and calibration poisoning for the CIFAR10 dataset when using a pretrained ResNet18 model (3) compared to training from scratch (2). We observe that reliability increases significantly when compared to the reliability of a model that was randomly initialized and trained from scratch. Notably, this means that improving the classifier's performance under partitioning of the training data will also improve the reliability of prediction sets and thus provides an interesting direction for future work, e.g. by applying recent improvements in the certifiable robustness literature (Wang et al., 2022; Rezaei et al., 2023). We also believe that studying poisoning in fine-tuning settings is a promising future research direction, especially given the proliferation of pretrained models in practice.

Finally, we provide additional results for our reliable prediction sets for the following evaluation metrics: (1) the ratio of empty sets, (2) the ratio of full sets, (3) the singleton ratio (ratio of sets containing a single class), and (4) the singleton hit ratio (empirical coverage of singleton prediction sets). Figure 12 shows results for the CIFAR10 dataset, and Figure 13 shows results for the SVHN dataset, for all three evaluation settings (a-c) described in Section 5.

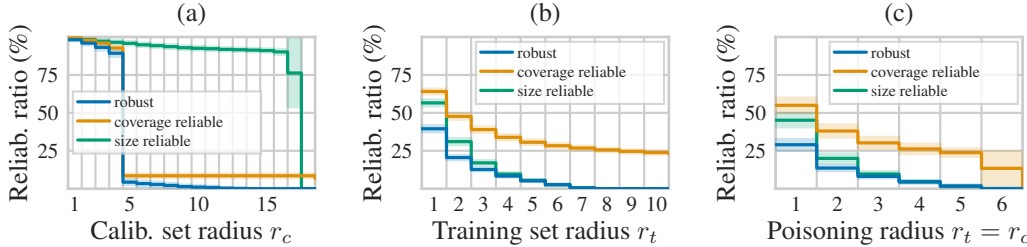

Figure 5: SVHN: Worst-case reliability guarantees across three scenarios: (a) poisoning of the calibration data, (b) poisoning of the training data, and (c) poisoning of both datasets.

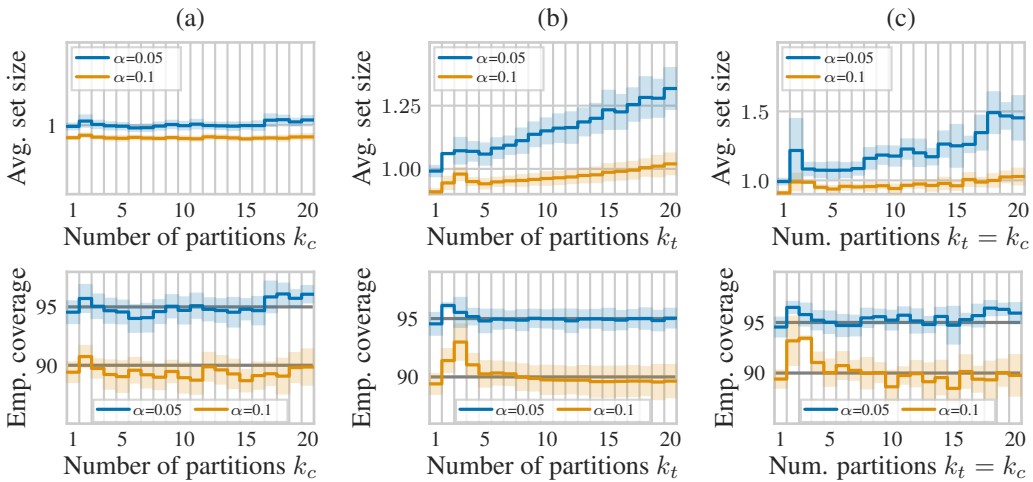

Figure 6: Avg. size and empirical coverage in all three settings (a–c) on the SVHN dataset.

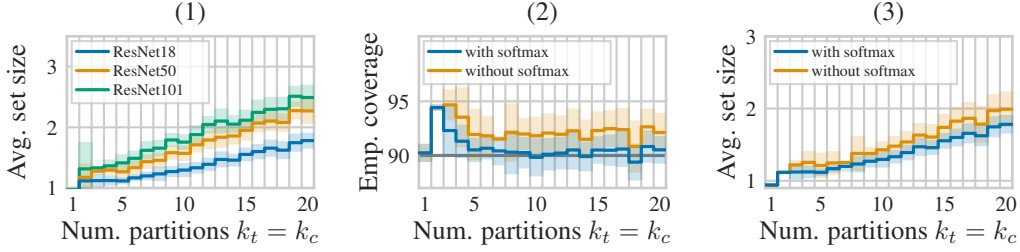

Figure 7: (1): Different models on CIFAR10. (2,3): Softmax ablation study for empirically justifying smoothing of the voting function (here with $\alpha = 0.1$) on CIFAR10.

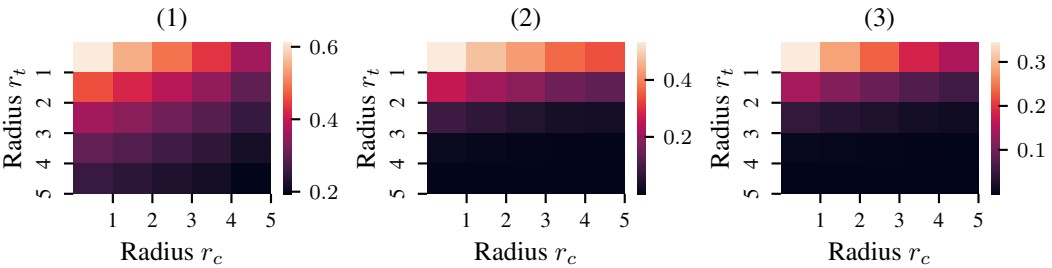

Figure 8: CIFAR10, $k_t = 100$, $k_c = 40$, $\alpha = 0.1$, (1): Coverage reliability ratio, (2): Size reliability ratio, (3): Robustness ratio (with with all radii combinations).

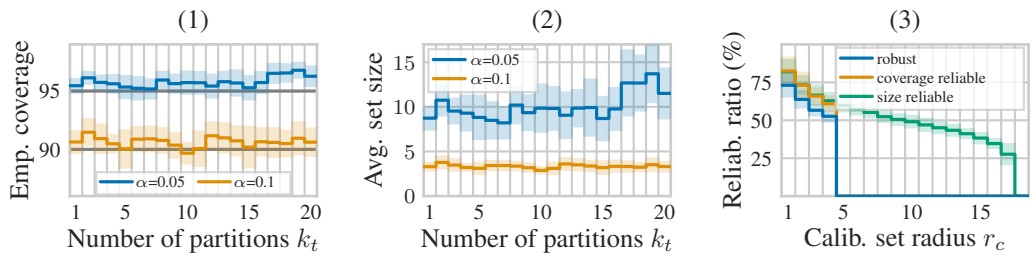

Figure 9: Additional results for ResNet18 on CIFAR100 under calibration poisoning: (1): Empirical coverage, (2): Average set size, (3): Reliability guarantees.

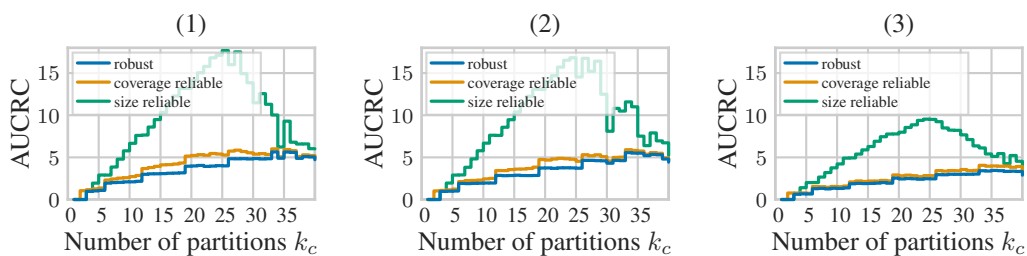

Figure 10: Relationship between reliability and the number of partitions under calibration poisoning for ResNet18 on (1): SVHN, (2): CIFAR10, and (3): CIFAR100. Here, AUCRC stands for the area under the certifiable reliability curve of the corresponding reliability types.

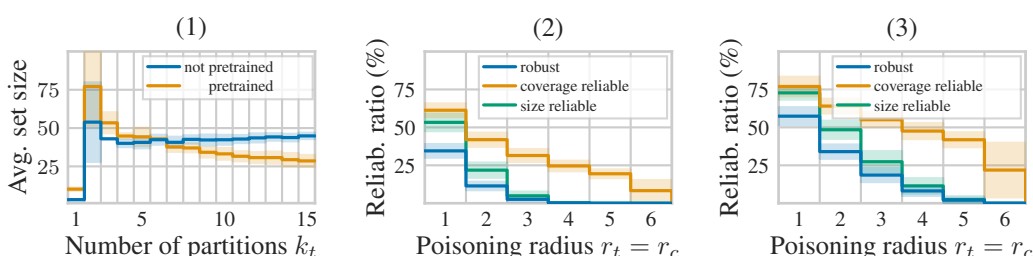

Figure 11: (1): Average set size using the smoothed score function on CIFAR100, comparing pretrained vs. non-pretrained models. (2-3): Reliability certificates under both training and calibration poisoning on CIFAR10, comparing reliability of randomly initialized ResNet18 models (2) – (as in Figure 2 c) – against the reliability of pretrained models that are only fine-tuned on CIFAR10 (3).

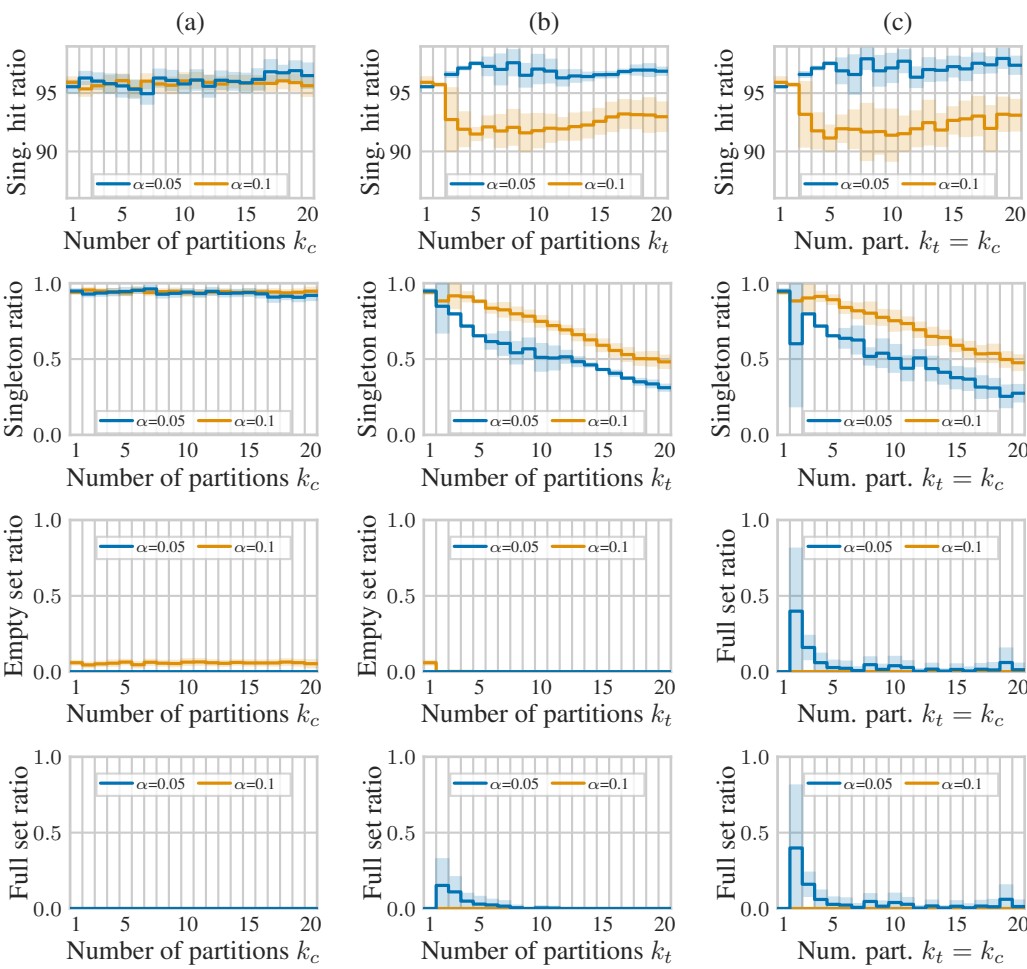

Figure 12: Various metrics for RPS in the three experiment settings (a-c) for ResNet18 on CIFAR10.

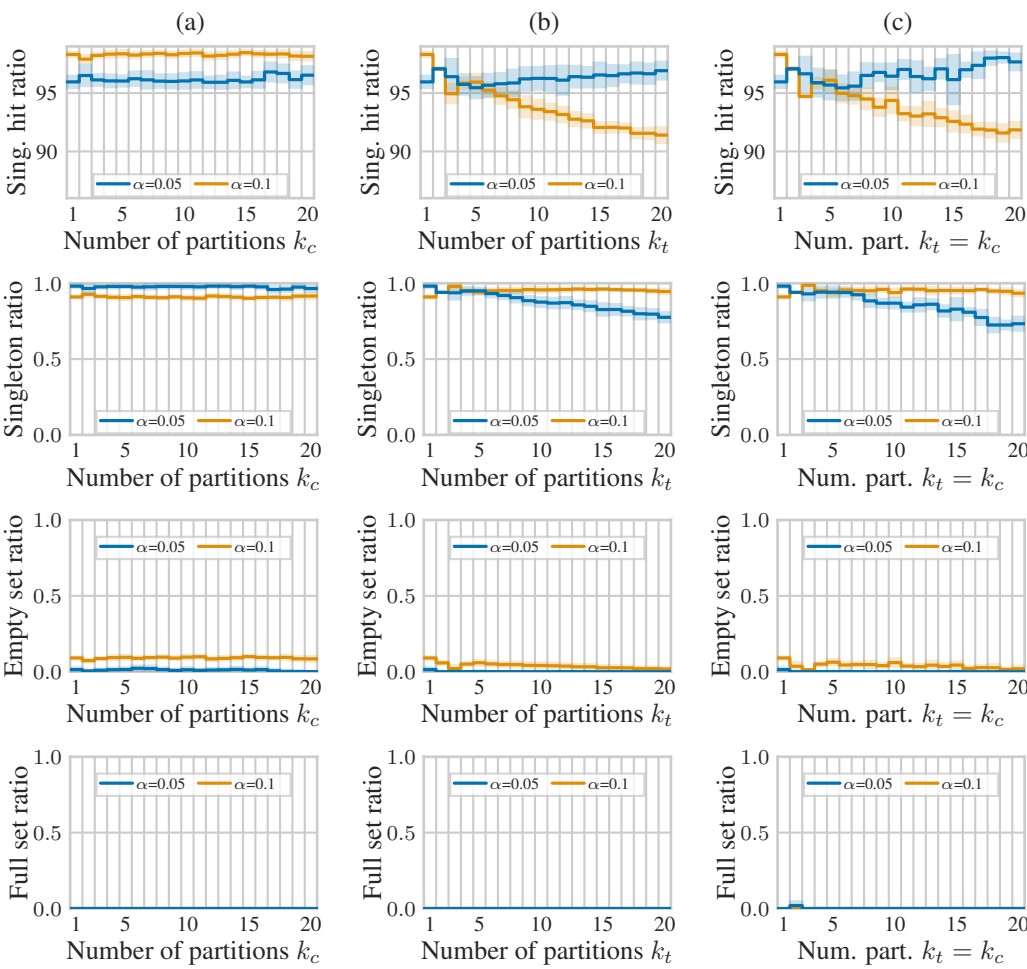

Figure 13: Various metrics for RPS in the three experiment settings (a-c) for ResNet18 on SVHN.

## B.1 Study of Statistical Efficiency of Majority Prediction Sets

Empirical coverage is a random variable following a Beta distribution concentrated around the nominal coverage level (Vovk, 2013). Here we conduct additional experiments to provide empirical evidence that the majority prediction sets are again closely concentrated around the nominal coverage level. We train a ResNet18 classifier on CIFAR10 and calibrate homogeneous prediction sets (Sadinle et al., 2019) for the nominal coverage level of $1 - \alpha = 0.9$. Then we compute the smaller prediction sets on $k_c = 10$ and $k_c = 20$ partitions, respectively. Finally, we merge the smaller prediction sets using the majority voting scheme described in Section 5. We repeat this process 100 times and compute the empirical coverage for each run. We then plot the empirical coverage for each run in Figure 14. As we partition the calibration set, the concentration naturally decreases for smaller prediction sets, leading to overcoverage. Interestingly, we observe that the majority prediction sets are again closely concentrated around the nominal coverage level, providing empirical evidence that majority prediction sets are statistically efficient.

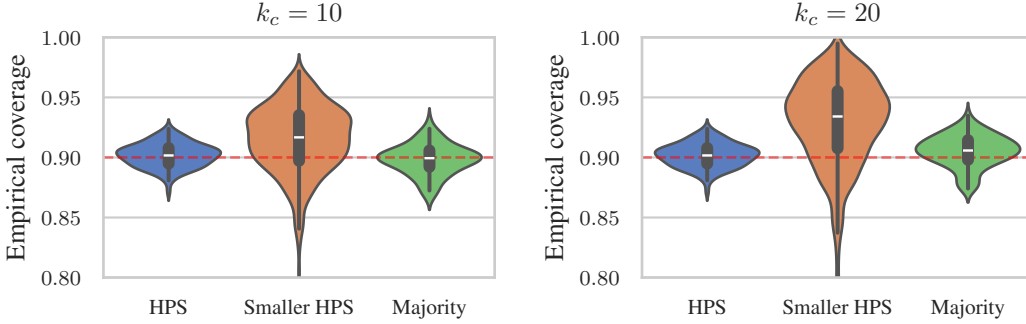

Figure 14: Concentration of empirical coverage for vanilla HPS, HPS for each partition, and the majority prediction sets. We use $k_c = 10$ partitions for the left plot, and $k_c = 20$ partitions for the right plot. Majority prediction sets are again closely concentrated around the nominal coverage level, providing empirical evidence that majority prediction sets are statistically efficient.

## C Reproducible Prediction Sets

There are conformal score functions that rely on random variables, e.g. for ensuring exact coverage by breaking ties. For example, adaptive prediction sets (APS) sum over class probabilities of classes with probability at least $f_y(x)$: $s(x, u, y) = -(\sum_{i=1}^{K} f_i(x) \mathbb{1}[f_i(x) > f_y(x)] + u f_y(x))$, where $f$ is a soft classifier and $u \in [0, 1]$ a uniform random variable (Romano et al., 2020). Conformal prediction sets are then formed by $\mathcal{C}(x_{n+1}) = \{s(x_{n+1}, u_{n+1}, y) \geq \tau\}$, where $u_{n+1} \in [0, 1]$.

Although randomization at inference changes prediction sets by at most one class (Angelopoulos et al., 2021), generating different prediction sets for the same input may not be desirable from a reliability standpoint and violates desideratum **V**. For example, a differential diagnosis for two patients with identical health parameters should not yield different results. As a solution, we propose *reproducible score functions*: Instead of drawing from a random variable, we propose to compute pseudorandom numbers by hashing the sum of the image's pixel values and initialize the pseudorandom number with the hash. We found that this is enough to break ties in practice while ensuring determinism required by our reliability guarantees. Note that our score functions with pseudorandomization are valid since they maintain exchangeability as the numbers depend solely on the data itself (not its position in the dataset or other datapoints).

We analyze this in Figure 15 (with $\alpha=0.05$): Without randomization APS results in large set sizes (2) despite tight coverage (1). Randomization leads to smaller sets but also compromises reproducibility. With pseudo-randomization, APS is reproducible with tight coverage and small prediction sets.

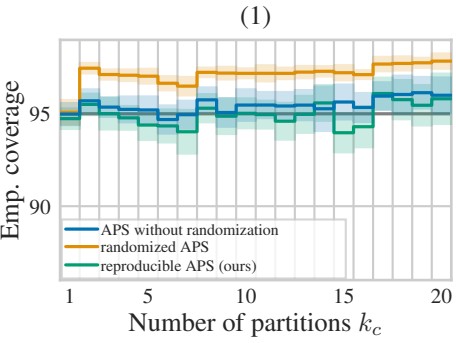 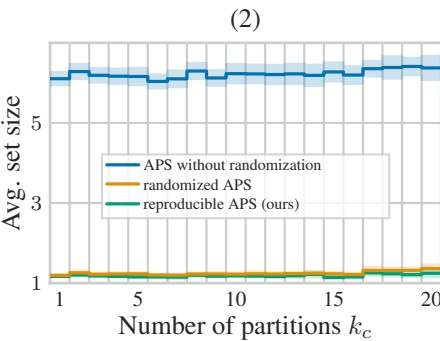

Figure 15: Comparing APS without and with randomization against our reproducible version.

## D  PROOFS FOR RELIABLE PREDICTION SETS (SECTION 5)

Recall the definition of the majority prediction sets from Section 5:

$$\mathcal{C}^M(x_{n+1}) = \left\{ y : \sum_{i=1}^{k_c} \mathbb{1}\{y \in \mathcal{C}_i(x_{n+1})\} > \hat{\tau}(\alpha) \right\}$$

with $\hat{\tau}(\alpha) = \max\{x \in [k_c] : F(x) \le \alpha\}$, where $F$ is the CDF of the Binomial distribution $\text{Bin}(k_c, 1 - \alpha)$ and $[k_c] = \{0, \dots, k_c\}$. In the following, denote $\hat{\tau} \triangleq \hat{\tau}(\alpha)$ for simplicity.

**Theorem 2.** *Given any conformal score function $s$ and test sample $(x_{n+1}, y_{n+1}) \in \mathcal{D}_{test}$ exchangeable with $\mathcal{D}_{calib}$, the majority prediction set (Equation 3) – constructed from sets calibrated on disjoint partitions – achieves marginal coverage on clean data:* $\Pr[y_{n+1} \in \mathcal{C}^M(x_{n+1})] \ge 1 - \alpha.$

*Proof.* Define the event $\phi_i \triangleq \mathbb{1}\{y_{n+1} \in \mathcal{C}_i(x_{n+1})\}$. The main argument is that the events $\phi_i$ and $\phi_j$ are independent for $i \ne j$. This holds since the prediction sets $\mathcal{C}_i$ and $\mathcal{C}_j$ are constructed on disjoint partitions of a calibration set of independent datapoints and are thus independent. This allow us to use an argument of Gasparin & Ramdas (2024) about majority voting in the context of uncertainty sets: Specifically, $\phi_i$ are independent Bernoulli random variables with $p_i = \Pr[\phi_i = 1] \ge 1 - \alpha$ by construction. Further define the random variable

$$S_{k_c} = \sum_{i=1}^{k_c} \mathbb{1}\{y \in \mathcal{C}_i(x_{n+1})\}.$$

First consider the special case $p_i = 1 - \alpha$ for all $i$. Then $S_{k_c}$ is a Binomial random variable, $S_{k_c} \sim \text{Bin}(k_c, 1 - \alpha)$. Thus we have:

$$\Pr[y_{n+1} \in \mathcal{C}^M(x_{n+1})] = \Pr\left[ \sum_{i=1}^{k_c} \mathbb{1}\{y_{n+1} \in \mathcal{C}_i(x_{n+1})\} > \hat{\tau} \right] \tag{5}$$

$$= \Pr[S_{k_c} > \hat{\tau}] \tag{6}$$

$$= 1 - \Pr[S_{k_c} \le \hat{\tau}] \tag{7}$$

$$= 1 - \underbrace{F(\hat{\tau})}_{\le \alpha} \tag{8}$$

$$\ge 1 - \alpha \tag{9}$$

In the general case of $p_i \ge 1 - \alpha$ we have that $S_{k_c}$ is distributed as a Poisson binomial random variable, $S_{k_c} \sim PB(k_c, [p_1, \dots, p_{k_c}])$. However, it holds that $\prod_{i=1}^{k_c} p_i > (1 - \alpha)^{k_c}$, which implies that the Poisson binomial distribution is stochastically larger than the Binomial distribution (Boland et al., 2002; Tang & Tang, 2023). See (Gasparin & Ramdas, 2024) for details. Intuitively, this means that the probability $\Pr[S_{k_c} > \hat{\tau}]$ can only increase in the general case where $p_i \ge 1 - \alpha$. $\square$

**Extended discussion.** In the context of uncertainty sets, Gasparin & Ramdas (2024) proved that merging independent uncertainty sets via majority voting can retain their $1-\alpha$ marginal coverage in general. However, note that their construction actually violates the marginal coverage guarantee: They define the threshold in their majority set as $q = \sup\{x \in \mathbb{R} : F(x) \leq \alpha\}$ (in contrast to our threshold $\hat{\tau} = \max\{x \in [k_c] : F(x) \leq \alpha\}$). The problem with their construction is that $F(q) > \alpha$ due to the definition of the supremum and the definition of the binomial CDF:

$$F(x; k_c, 1-\alpha) = \sum_{i=0}^{\lfloor x \rfloor} \binom{k_c}{i}(1-\alpha)^i \alpha^{k_c-i}$$

This means $q = \hat{\tau} + 1$, which leads to smaller majority sets that violate the coverage guarantee since $\Pr[S_{k_c} > q] < 1-\alpha$ due to $F(q) > \alpha$. Beyond that, we propose a method for improving the reliability of conformal prediction against worst-case poisoning attacks (by partitioning the calibration set), and derive reliability guarantees.

**Minimal calibration set size for majority prediction sets.** Recall that desideratum **VI** (Section 4) requires that the reliability of prediction sets must increase for larger calibration sets. Our majority prediction sets fulfill this desideratum by construction since increasing the number of partitions $k_c$ will decrease the influence of datapoints. However, we need enough data to construct prediction sets: due to the finite-sample correction, the calibration partitions cannot become arbitrarily small. This naturally bounds the number of partitions $k_c$ for a fixed calibration size $n$ (and thus reliability):

**Proposition 1.** *Let $i^* = \arg\min_{i \in \{1,\ldots,k_c\}} |P_i^c|$ denote the partition of smallest size. Given coverage probability $1-\alpha$, we can construct prediction sets for partition $i$ provided that $|P_{i^*}^c| \geq (\frac{1}{\alpha} - 1)$.*

*Proof.* In general, constructing prediction sets given $n$ calibration points requires, due to the finite sample correction, that $0 \leq \alpha_n = \lfloor \alpha(n+1) - 1 \rfloor / n$ holds (otherwise we cannot compute quantiles). We have: $0 \leq \lfloor \alpha(n+1) - 1 \rfloor / n \Leftrightarrow 1 \leq \lfloor \alpha(n+1) \rfloor \overset{(1)}{\Leftrightarrow} 1 \leq \alpha(n+1) \Leftrightarrow \frac{1}{\alpha} - 1 \leq n$, where (1) holds since the l.h.s. is a natural number.

Thus we need $n \geq \frac{1}{\alpha} - 1$ datapoints in our calibration set. If we partition the calibration data into $k$ equally-sizes subsets of size $\frac{n}{k}$, then we need at least $n \geq k\left(\frac{1}{\alpha} - 1\right)$ datapoints. If the partitions are not equally-sized, then we require for the smallest partition $i^*$ that $|P_{i^*}^c| \geq (\frac{1}{\alpha} - 1)$ holds. □

## E    PROOFS FOR RELIABILITY CERTIFICATES (SECTION 6)

---

**Algorithm 3** Greedy algorithm for upper bounding the smoothed score function $s$

**Input:** Score function $s$, $x$, $y$, $k_t$, $r_t$
1: $\pi = [\pi_1(x), \ldots, \pi_K(x)]$
2: **for** $i = 1$ **to** $r_t$ **do**
3:     $\hat{y} = \arg\max_{\hat{y} \neq y} \pi_{\hat{y}}$
4:     $\pi_{\hat{y}} \leftarrow \min(\max(\pi_{\hat{y}} - 1/k_t, 0), 1)$
5:     $\pi_y \leftarrow \min(\max(\pi_y + 1/k_t, 0), 1)$
**Output:** $\overline{s}(x, y) = e^{\pi_y} / (\sum_{i=1}^{K} e^{\pi_i})$

---

**Algorithm 4** Greedy algorithm for lower bounding the smoothed score function $s$

**Input:** Score function $s$, $x$, $y$, $k_t$, $r_t$
1: $\pi = [\pi_1(x), \ldots, \pi_K(x)]$
2: **for** $i = 1$ **to** $r_t$ **do**
3:     **if** $\pi_y = 0$ **then**
4:         $y' \leftarrow y$
5:     **else**
6:         $y' \leftarrow \arg\min_{y':\pi_{y'}>0} \pi_{y'}$
7:     $\pi_{y'} \leftarrow \min(\max(\pi_{y'} - 1/k_t, 0), 1)$
8:     $\hat{y} = \arg\max_{\hat{y} \neq y} \pi_{\hat{y}}$
9:     $\pi_{\hat{y}} \leftarrow \min(\max(\pi_{\hat{y}} + 1/k_t, 0), 1)$
**Output:** $\underline{s}(x, y) = e^{\pi_y} / (\sum_{i=1}^{K} e^{\pi_i})$

---

Let $\mathcal{C}(x_{n+1}) = \{y \in \mathcal{Y} : s(x_{n+1}, y) \geq \tau\}$ be a prediction set for a new test point $x_{n+1}$ derived using conformal prediction (Section 3) under the clean dataset $\mathcal{D}_l$ and with the smoothed score function $s$.

**Lemma 2.** *We can upper bound the score function for any $\tilde{\mathcal{D}}_l \in B_{r_t,r_c}(\mathcal{D}_l)$ as follows:*

$$\overline{s}(x,y) = \max_{\substack{0 \leq \pi_i \leq 1 \\ \Delta_i \in \{0, \pm\frac{1}{k_t}, \dots, \pm\frac{r_t}{k_t}\} \\ \sum_{i=1}^K \Delta_i = 0}} \frac{e^{\pi_y}}{\sum_{i=1}^K e^{\pi_i}} \quad \text{with} \quad \pi = [\pi_1(x) + \Delta_1, \dots, \pi_K(x) + \Delta_K] \quad (10)$$

*Proof.* Since adversaries can insert or delete at most $r_t$ datapoints, we know at most $r_t$ training partitions can be affected in the worst-case. Thus at most $r_t$ of $k_t$ classifiers change their prediction. $\square$

This holds analogously for the lower bound on the smoothed score function:

$$\underline{s}(x,y) = \min_{\substack{0 \leq \pi_i \leq 1 \\ \Delta_i \in \{0, \pm\frac{1}{k_t}, \dots, \pm\frac{r_t}{k_t}\} \\ \sum_{i=1}^K \Delta_i = 0}} \frac{e^{\pi_y}}{\sum_{i=1}^K e^{\pi_i}} \quad \text{with} \quad \pi = [\pi_1(x) + \Delta_1, \dots, \pi_K(x) + \Delta_K] \quad (11)$$

**Proposition 2.** *Algorithm 3 and Algorithm 4 solve the discrete optimization problems in Equation 10 and Equation 11, respectively. The optimal solutions represent the worst-case bounds on the score function $s$ under any poisoned dataset $\tilde{\mathcal{D}}_l \in B_{r_t,r_c}(\mathcal{D}_l)$.*

*Proof.* In the worst-case, the adversary controls at most $r_t$ partitions, which means the adversary controls the predictions of at most $r_t$ classifiers and can consequently change at most $\frac{r_t}{k_t}$ probability mass in the vote-distribution $\pi$ over classes $\mathcal{Y}$, which is exactly what the two optimization problems model. We now distinguish between the two cases:

- For the upper bound $\overline{s}(x,y)$, the worst-case adversary redistributes the probability mass from the classes with the largest probability masses to the target class $y$, which is the worst-case upper bound since it maximizes the numerator and minimizes the denominator.

- For the lower bound $\underline{s}(x,y)$, the worst-case adversary redistributes the probability mass from the target class to the class with the largest probability mass. If the target class has 0 remaining probability mass, then the probability from the smallest class with probability mass larger than 0 is redistributed to the class with most of the probability mass. This is the worst-case lower bound since it minimizes the numerator and maximizes the denominator.

The argument holds since the space we are optimizing over is discrete, $\pi_i(x) \in \{0, \frac{1}{k_t}, \dots, \frac{k_t-1}{k_t}, 1\}$. Both worst-cases are exactly what the Algorithms in Algorithm 3 and Algorithm 4 compute. $\square$

Clearly, both greedy algorithms need $r_t$ iterations to terminate (due to the for loop).

**Theorem 3.** *Given $r_c$=0, the conformal prediction set $\tilde{\mathcal{C}}(x_{n+1})$ derived with the smoothed score function under any poisoned dataset $\tilde{\mathcal{D}}_l \in B_{r_t,r_c}(\mathcal{D}_l)$ is coverage reliable, i.e. $\tilde{\mathcal{C}}(x_{n+1}) \supseteq \mathcal{C}(x_{n+1})$, if $\underline{s}(x,\underline{y}) \geq \overline{\tau}$, and size reliable, i.e. $\tilde{\mathcal{C}}(x_{n+1}) \subseteq \mathcal{C}(x_{n+1})$, if $\overline{s}(x,\overline{y}) < \underline{\tau}$.*

*Proof.* We consider training set poisoning. In the worst case, adversaries perturb at most $r_t$ training partitions. Thus, the worst-case quantiles are given by:

$$\underline{\tau} = \text{Quant}(\alpha_n; \{\underline{z_i}\}_{i=1}^n) \qquad \overline{\tau} = \text{Quant}(\alpha_n; \{\overline{z_i}\}_{i=1}^n)$$

where $\underline{z_i} = \underline{s}(x_i,y_i)$ and $\overline{z_i} = \overline{s}(x_i,y_i)$ are lower and upper bounds on the scores for all points $(x_i,y_i) \in \mathcal{D}_{calib}$ in the calibration set. We treat coverage and size reliability separately:

*Coverage reliability*: We consider a prediction set as coverage reliable if no class can be removed from the set. If adversaries cannot remove the "weakest" class $\underline{y}$ with the fewest votes $\pi_y$ among all classes $y \in \mathcal{C}(x_{n+1})$ from the prediction set, then adversaries also cannot remove any other class since they would need even more adversarial budget. In the worst-case, the lowest score for sample $x$ and class $y$ is given by $\underline{s}(x,y)$. Thus, if this lowest score is still larger than or equal to the worst-case quantile $\overline{\tau}$, then the weakest class cannot be removed and the prediction set is coverage reliable.

*Size reliability*: We consider a prediction set as size reliable if no class can be added to the set. If adversaries cannot add the "strongest" class $\overline{y}$ with the most votes $\pi_y$ among all classes $y \notin \mathcal{C}(x_{n+1})$ into the prediction set, then adversaries also cannot add any other class since they would need even more adversarial budget. In the worst-case, the largest score for sample $x$ and class $\overline{y}$ is given by $\overline{s}(x, \overline{y})$. Thus, if this largest score is still smaller than the worst-case quantile $\underline{\tau}$, then the strongest class cannot be added and the prediction set is size reliable. $\square$

**Theorem 4.** *Given $r_t=0$ and deterministic score function $s$, the majority prediction set $\tilde{\mathcal{C}}^M(x_{n+1})$ derived under any dataset $\tilde{\mathcal{D}}_l \in B_{r_t,r_c}(\mathcal{D}_l)$ is coverage reliable, i.e. $\tilde{\mathcal{C}}^M(x_{n+1}) \supseteq \mathcal{C}^M(x_{n+1})$, if $\underline{m} - r_c > \hat{\tau}(\alpha)$, and size reliable, i.e. $\tilde{\mathcal{C}}^M(x_{n+1}) \subseteq \mathcal{C}^M(x_{n+1})$, if $\overline{m} + r_c \leq \hat{\tau}(\alpha)$, provided that the smallest calibration partition $i^*$ is large enough $|P_{i^*}^c| - r_c \geq (\frac{1}{\alpha} - 1)$.*

*Proof. Coverage reliability:* In the worst-case, adversaries control at most $r_c$ calibration partitions and thus can change the support $\sum_{i=1}^{k_c} \mathbb{1}\{y \in \mathcal{C}_i(x_{n+1})\}$ for classes $y \in \mathcal{C}^M(x_{n+1})$ by at most $r_c$. If removing $r_c$ from the support $\underline{m}$ of the class with the least support is not enough to remove the class, $\underline{m} - r_c > \hat{\tau}$, then the majority prediction set remains coverage reliable.

*Size reliability:* In the worst-case, adversaries control at most $r_c$ calibration partitions and thus can change the support $\sum_{i=1}^{k_c} \mathbb{1}\{y \in \mathcal{C}_i(x_{n+1})\}$ for classes $y \notin \mathcal{C}^M(x_{n+1})$ by at most $r_c$. If adding $r_c$ to the support $\overline{m}$ of the class with the most support is not enough to add the class, $\overline{m} + r_c \leq \hat{\tau}$, then the majority prediction set remains size reliable.

This only holds if the smallest calibration partition $i^*$ is large enough $|P_{i^*}^c| - r_c \geq (\frac{1}{\alpha} - 1)$ under an attack, otherwise the adversary could prevent us from computing the majority prediction set in the first place by deleting datapoints from partition $i^*$. $\square$

**Theorem 5.** *Let $\beta_y$ denote the number of prediction sets $\mathcal{C}_i \in \{\mathcal{C}_1, \dots, \mathcal{C}_{k_c}\}$ for which we can guarantee $y \in \mathcal{C}_i$ under $r_t$ poisoned training datapoints. If $\beta_y - r_c > \hat{\tau}(\alpha)$ for all $y \in \mathcal{C}^M(x_{n+1})$ then the majority prediction set is coverage reliable under any dataset $\tilde{\mathcal{D}}_l \in B_{r_t,r_c}(\mathcal{D}_l)$, provided that the smallest calibration partition $i^*$ is large enough $|P_{i^*}^c| - r_c \geq (\frac{1}{\alpha} - 1)$.*

*Proof.* If there is a single class $y \in \mathcal{C}^M(x_{n+1})$ for which we cannot guarantee that more than $\hat{\tau}$ prediction sets contain $y$ under $r_t$ poisoned training and $r_c$ poisoned calibration points, then the majority prediction set is not coverage reliable in the worst-case. Thus showing $\beta_y - r_c > \hat{\tau}$ for all $y \in \mathcal{C}^M(x_{n+1})$ as explained in the main text is a sufficient condition for coverage reliability. $\square$

**Theorem 6.** *Let $\gamma_y$ denote the number of prediction sets $\mathcal{C}_i \in \{\mathcal{C}_1, \dots, \mathcal{C}_{k_c}\}$ for which we can guarantee $y \notin \mathcal{C}_i$ under $r_t$ poisoned training datapoints. If $k_c - \gamma_y + r_c \leq \hat{\tau}(\alpha)$ for all $y \notin \mathcal{C}^M(x_{n+1})$ then the majority prediction set is size reliable under any dataset $\tilde{\mathcal{D}}_l \in B_{r_t,r_c}(\mathcal{D}_l)$, provided that the smallest calibration partition $i^*$ is large enough $|P_{i^*}^c| - r_c \geq (\frac{1}{\alpha} - 1)$.*

*Proof.* If there is a single class $y \notin \mathcal{C}^M(x_{n+1})$ for which we cannot guarantee that less than (or equal to) $\hat{\tau}$ prediction sets do not contain $y$ under $r_t$ poisoned training points and $r_c$ poisoned calibration points, then the majority prediction set is not size reliable in the worst-case. If $\gamma_y$ denotes the number of prediction sets for which we can guarantee $y \notin \mathcal{C}_i$ under $r_t$ poisoned training datapoints, then $k_c - \gamma_y$ is the number of prediction sets for which we cannot guarantee $y \notin \mathcal{C}_i$ (this entails the number of prediction sets with $y \in \mathcal{C}_i$). Under consideration of additional calibration poisoning, we cannot guarantee $y \notin \mathcal{C}_i$ for $k_c - \gamma_y + r_c$ prediction sets. In the worst case, $k_c - \gamma_y + r_c$ prediction sets will contain the class. In other words, $k_c - \gamma_y + r_c \leq \tau$ for all $y \notin \mathcal{C}^M(x_{n+1})$ is a sufficient condition for size reliability. $\square$

**Computational complexity.** For the training-poisoning certificates we have to compute the algorithm for all $n$ calibration points. Assuming we recompute the argmax every time, the certificates can be computed in $\mathcal{O}(nr_tK)$ (where $K$ is the number of classes). Regarding our calibration-poisoning certificates, the terms $\underline{m}, \overline{m}$ can be computed efficiently in $\mathcal{O}(Kk_c)$ steps. To compute guarantees in the general case, we mainly need to compute the terms $\beta_y$ and $\gamma_y$ for all $K$ classes $y \in \mathcal{Y}$. This involves computing the worst-case quantiles ($\mathcal{O}(\frac{n}{k_c}r_tK)$) for each prediction set $k_c$ and the worst-case score ($\mathcal{O}(r_tK)$). Thus overall the guarantees can be computed in $\mathcal{O}(Kk_c(\frac{n}{k_c}r_tK)) = \mathcal{O}(K^2nr_t)$.

