# OpenReview forum: "Provably Reliable Conformal Prediction Sets in the Presence of Data Poisoning"
_ICLR.cc/2025/Conference — ICLR 2025 Spotlight_

### Official Review · Reviewer_AjMN · 2024-10-28

**Soundness:** 4
**Presentation:** 3
**Contribution:** 4
**Rating:** 8
**Confidence:** 4

**Summary:**

This work studies robustness of conformal prediction to poisoning in a threat model that allows insertion and deletion which is a limitation of the other studies e.g. zargarbashi et al, 2024. For training the authors propose to divide the training data into $k_t$ disjoint sets so that by definition any poisoning will at worst break one model. This robustness approach was previously studied in image classification as well. The conformity score function is the softmax over the votes of each classifier on each class. While the conformal guarantee is safe w.r.t. poisoning in the training process, calibration data can be poisoned as well. The authors remedy that by a similar approach of dividing the calibration set, and producing prediction sets from each calibration group, and running a majority vote-based decision on the final output.

Overall I don’t count section 5.1 as a novel approach tailored to conformal prediction. However it is an smart approach to reduce the poisoning effect on set size. However showing that Section 5.2 is a valid approach is an important contribution.

**Strengths:**

The other works designed to make CP robust to poisoning are not considering insertion and deletion in their threat model. This makes this work unique. Also their approach however repeated from the poisoning literature, still is novel to be adapted to conformal prediction framework.

The score function designed by the authors is interesting and novel. It also can have interesting properties beyond the score of robustness. The interesting outcome of this is that it enables them to provide the guarantees on the set size which is novel.

The paper is theoretically solid. I think the theorems are well defined, and the results are quite far from obvious. Therefore I see this as a strong work.

The writing is also in a good shape; the story of the paper is easy to follow. However a strong recommendation is to add a discussion on differences between normal conformal coverage guarantee and coverage reliability.

**Weaknesses:**

**Writing.**  I expect the threat model to be mentioned more clearly in the introduction. Until the related work I did not understood that the authors are addressing a poisoning model different from zargarbashi et al 2024. I strongly suggest the authors to mention that in their threat model addition and deletion of data points is allowed which is a limitation of zargarbashi et al 2024 either in the introduction or in the corresponding section in related works.

**Wrong background.** The definition of the conformal prediction in Theorem 1 is not correct. Either you define the scores as capturing agreement between the data and the label (e.g. directly using softmax) then the for test points you add labels with scores above the threshold and the threshold is the $\alpha_n = \lfloor\alpha(n+1)-1\rfloor$ or the quantile is as authors specified in line 135, where the score should be non-conformity $1 - f_y(x)$ and the labels with score lower that the threshold should be picked.

**Unnecessary, and unclear desiderata.** Although I agree on desiderata no. 1, 2, and 3, I still think that computational efficiency is not a necessity. Also if no. 4 means that there is an strict increasing requirement, then it contradicts the assumption of adding datapoints can be invariant to the final result. Other than that what is the concrete definition of providing small prediction sets? Is it in comparison to the same model trained on clean data? or is it in comparison to vanilla setup?

**Discrete score.**  Although the score is float, it is discrete with fixed number of possibilities. However working in practice there are corner cases where ties can become problematic. You need to add a random noise to the score in Eq. 2 which I think does not change that much in the efficiency of the output but simply avoids these small chances of failure.

**Study of statistical efficiency.** The empirical coverage is not always $1 - \alpha$. This is on it’s own a random variable sampled from a Beta distribution concentrated around the nominal guarantee. This concentration is increased by the number of calibration points. The first question that is not addressed here is that how breaking the calibration set to smaller calibration sets influence this concentration? For sure it should decrease due to smaller calibration sets, while some of this can be remedied by the majority prediction set. Do you have any intuition on how this is affected? At least I expect a discussion on the lowerbound of this in the limitations of the work.

**Typos.** Sentence at the end of line 250 is grammatically wrong. Typos are seen in lines 349 (title).

**Suggestions.** The intuition behind the proof of Theorem 2. is non-trivial. I would recommend the authors to add a short sketch of proof or an intuition on what is deciding the parameter of majority vote set in the manuscript.

**Questions:**

1. The definition of $\hat{\tau}$ is unclear to me. Can you elaborate? What is this capturing? Why the iteration variable is called $x$? Does it have any correlation with covariates?
2. I can not understand why poisoning in training data can influence the coverage reliability. Since the coverage is only dependent to calibration set the guarantee should not break with a poisoned model (CP is guaranteed with black-box model). Then why Fig. 2-b shows a decay on coverage reliable. In general there are no comprehensive discussions on what coverage reliability and coverage guarantee have in difference. So from the conformal guarantee changes in the training data should have no influence in the “marginal” coverage of the prediction sets. However I think this reliability is defined per point. Is that so? In case yes please update the manuscript to draw this difference line in a more clear way.
3. I think the set-size reliability is misleading to show (at least as a solid line) for calibration poisoning. Under calibration poisoning the guarantee is already invalid and with an invalid guarantee by definition it is not important if the set sizes are small or not. Even in some cases expanding set sizes can be considered as an alert to the defender that the calibration set is invalid.
4. In Fig 3. I can not understand the jumps in the beginning of the x axis for empirical coverage? What is the reason behind that intuitively? I see a quick explanation in lines 473-475 but isn’t it due to the missing randomization in the score function?

If the points in questions and weaknesses are provided I would be happy to increase my score.

---

> ### Author Response · Authors · 2024-11-19
> **Response to Reviewer AjMN**
>
> Thank you for your review!
>
> **Regarding our threat model.** Thank you for acknowledging the novelty of our threat model for conformal prediction. Following your suggestion, we have clarified the threat model in the introduction (line 47) and point out limitations of the threat model presented in (Zargarbashi et al., 2024) more explicitly within the related work section (line 98).
>
> **Regarding the background**. We have updated $\alpha_n$ in line 135 to the correct threshold when scores capture the agreement between data and label. Please note this minor correction in the background does not affect any of our theoretical or empirical results. Thank you for bringing this to our attention.
>
> **Regarding desiderata for reliable conformal prediction.** Please note that the desiderata serve as foundation for how we envision reliable conformal prediction, focusing not only on the reliability of prediction sets but also on their practical relevance. This includes maintaining small set sizes (comparable to prediction sets without reliability guarantees), and ensuring computational efficiency in their construction, making them applicable in real-world settings. Regarding desideratum IV, it is not intended as a strict requirement for prediction sets, it rather ensures that algorithms have the flexibility to increase reliability when practical risk increases with more data. In response to your feedback, we have added clarifications to desiderata II, IV, and V, expressing more clearly that reliable conformal prediction requires efficient and flexible algorithms. Thank you for pointing out opportunities for further clarification.
>
> **Regarding conformal score ties.** As you correctly pointed out, one has to take care of ties in practice for tighter coverage. To address this issue, we employ a deterministic softmax function to smooth the voting function $\pi_y(x)$. As demonstrated in our ablation study (Figure 4), this approach effectively keeps the prediction set size low, resulting in tighter empirical coverage. Please note that introducing additional random noise to the score function would compromise provable reliability guarantees under training poisoning. Interestingly, this also means that most existing score functions in the literature lack reliability by definition (see our discussion in Appendix D). In contrast, our first approach addresses this issue while ensuring stability of prediction sets at the same time. Thank you for bringing this up, this discussion highlights again the inherent challenges of providing *provably* reliable prediction sets.
>
> **Regarding a study of statistical efficiency.** As you correctly pointed out, the empirical coverage is a random variable following a Beta distribution concentrated around the nominal coverage level. As we partition the calibration set, this concentration naturally decreases for the smaller prediction sets. In response to your comment, we have conducted additional experiments (Appendix C, Figure 13) to provide empirical evidence that the majority prediction sets are again closely concentrated around the nominal coverage level. We have also added the discussion to the main text (line 469-470).
>
> **Typos (L250, L349).** We also fixed the typos, thank you for pointing this out.
>
> ### References
>
> Soroush H. Zargarbashi, Mohammad Sadegh Akhondzadeh, Aleksandar Bojchevski. Robust Yet Efficient Conformal Prediction Sets. ICML 2024.

---

> ### Author Response · Authors · 2024-11-19
> **Response to Reviewer AjMN**
>
> **Regarding the definition of $\hat{\tau}$ (Q1)**. The definition of $\hat{\tau}(\alpha)$ just corresponds to a quantile function, in this case the inverse of the CDF $F$ of the Binomial distribution $\text{Bin}(k_c, 1-\alpha)$. Intuitively, we select $\hat{\tau}$ such that the probability of a class $y$ being in at most $\hat{\tau}$ of $k_c$ prediction sets is at most $\alpha$, provided that $y$ is in each individual prediction set with probability at least $1-\alpha$ (which holds for $y_{n+1}$ since the smaller prediction sets are calibrated). Following your suggestion, we have added further intuition and clarifications about $\hat{\tau}$ along with a proof sketch for Theorem 2 to the manuscript.
>
> **Regarding coverage guarantee vs. reliability (Q2).** You are correct that while the coverage guarantee is a marginal guarantee over the entire data distribution, our definition of coverage reliability is point-wise, meaning it applies to individual prediction sets $\mathcal{C}(x)$. Please note that, unlike the coverage guarantee, point-wise reliability can indeed be affected by training poisoning. In response to your comment, we have updated the manuscript to clarify this distinction. Specifically, we address it directly when introducing our notion of reliability (line 161) and further elaborate on it in the discussion (lines 521–529). Thank you for helping us improve the clarity of our work.
>
> **Regarding set-size reliability in calibration poisoning (Q3).** As we previously clarified, our notion of reliability is point-wise, meaning it applies to individual prediction sets. We believe that demonstrating the point-wise size reliability of prediction sets is both sound and meaningful. Additionally, while expanding set sizes might serve as an empirical defense mechanism to flag invalid prediction sets, such defenses lack provable guarantees under worst-case adversarial attacks. In contrast, our approach ensures that prediction sets remain small while providing provable stability guarantees within a defined attack radius.
>
> **Regarding Figure 3 (Q4).** Intuitively, we observe initial jumps in empirical coverage and set size in Figure 3 when using only $k_t=2$ or $3$ classifiers, because a sufficient number of classifiers is required to achieve consistent majority vote consensus and reliability in practice. Please note that our analysis demonstrates that four classifiers are already sufficient to prevent such jumps. In response to your comment, we have revised the manuscript to clarify this observation (lines 471-475).
>
>
> We hope that we could address all your questions to your satisfaction. Please let us know if you have any additional comments or questions.

---

> > ### Comment · Reviewer_AjMN · 2024-11-20
> >
> > I am really thankful for addressing my concerns and adding experiments. In light of the response, and as the comments are addressed I am happy to increase my score.

---

### Official Review · Reviewer_HznX · 2024-11-04

**Soundness:** 3
**Presentation:** 3
**Contribution:** 3
**Rating:** 8
**Confidence:** 3

**Summary:**

This paper proposes reliable prediction sets (RPS) as an efficient method for constructing conformal prediction sets under training and calibration data poisoning. The proposed approach has two components: to improve reliability under training data poisoning, authors introduce smoothed score functions that reliably aggregate predictions from classifiers trained on distinct partitions of the training data; and to improve reliability under calibration data poisoning, authors calibrate multiple prediction sets, each calibrated on distinct subsets of the calibration data, and construct a majority prediction set. Authors propose a desiderata for reliable conformal prediction, and derive certificates for the reliability of prediction sets under worst-case data poisoning attacks. The paper presents empirical evaluation of the method on image classification tasks.

**Strengths:**

1. The paper is well-written, organized, and easy to follow for the most part.
2. The paper is technically sound. I appreciate the paper clearly lays out the desiderata for reliable conformal prediction sets under data poisoning in the beginning. The claims regarding RPS are well supported by theory and empirical evaluation.
3. The paper presents important results that can contribute toward improving the reliability of conformal prediction sets under data poisoning attacks in real scenarios.

**Weaknesses:**

1. It would help to show comparison with baselines in the empirical evaluation. I see there is some comparison with APS to understand the effect of randomization; however, it would be interesting to see worst-case reliability analysis comparison with existing conformal prediction methods, and discussing how RPS fares in the different desiderata.
2. Relationship of reliability with number of partitions: the paper briefly discusses how the number of partitions (and dataset size) will affect the reliability of prediction sets; however, I did not seem to find experiments that analyze this relationship. It would help to demonstrate this relationship in order to better understand the utility of prediction sets in different scenarios.
3. Section 6 can be made more readable and the notation becomes slightly hard to follow at times.



**Minor comments:**
1. L176: typo, Desideratum VI -> Desideratum IV
2. L300: equation reference should probably be (5)
3. L350: typo, coverave -> coverage

**Questions:**

Is there any reason why full experiments for CIFAR-100 and ImageNet were not included? I appreciate the paper mentions the limitation in terms of accuracy loss, and I understand if full training can be expensive but I still believe there is value in showing limited results here, especially under calibration poisoning. It will be good to understand performance and the various desiderata in these relatively harder datasets with larger number of classes.

---

> ### Author Response · Authors · 2024-11-19
> **Response to Reviewer HznX**
>
> Thank you for your review!
>
> **Regarding existing conformal prediction methods.** Thank you for acknowledging our contribution toward improving the reliability of conformal prediction sets under data poisoning. Please note that all existing conformal prediction methods do not provide any provable guarantees for reliability under our poisoning threat model. Given the development of first poisoning attacks against conformal prediction (Li et al., 2024), this discussion highlights again the need for developing *provably* reliable prediction sets.
>
> **Regarding the relationship between reliability vs. number of partitions.** Thank you for pointing out opportunities for further clarifications. Intuitively, a larger number of training partitions $k_t$ increases reliability against training poisoning, and a larger number of calibration partitions $k_c$ increases reliability against calibration poisoning. In response to your comment, we provide additional experiments and details in Appendix B (Figure 10) to demonstrate this relationship empirically.
>
> **Regarding large-scale experiments.** Please note that we present results on CIFAR-100 in Figure 4 (1) and include additional results regarding this dataset in Appendix B. In response to your comment, we provide additional plots for this dataset in Appendix B for completeness (Figure 9 and 10). Beyond this, we believe that our experiments on standard benchmarks commonly used in reliability research are sufficient to demonstrate the effectiveness of our method. Nevertheless, we are prepared to provide additional resource-intense training experiments in the final version of the manuscript.
>
> **Regarding typos.** We also fixed the typos (L176, L300, L350), thank you for bringing this to our attention.
>
> We hope that we could address all your questions to your satisfaction. Please let us know if you have any additional comments or questions.
>
> ### References
> Yangyi Li, Aobo Chen, Wei Qian, Chenxu Zhao, Divya Lidder, Mengdi Huai. Data Poisoning Attacks against Conformal Prediction. ICML 2024.

---

> > ### Comment · Reviewer_HznX · 2024-11-25
> >
> > I would like to thank the authors for their response. I maintain my score in favor of accepting the paper.

---

### Official Review · Reviewer_1V8P · 2024-11-05

**Soundness:** 3
**Presentation:** 3
**Contribution:** 3
**Rating:** 6
**Confidence:** 3

**Summary:**

They introduce a method called "reliable prediction sets" that constructs conformal prediction reliable sets under poisoning attacks. They consider both training and calibration data poisoning. In order to achieve robustness against training poisoning, they partition training data into different parts and train a classifier on each part, and then aggregate the predictions from these classifiers to derive a smooth conformal score function.

Furthermore, in order to be reliable against calibration poisoning, they calibrate multiple prediction sets on disjoint subsets of the calibration data and construct a majority prediction set that includes classes that are included in the majority of the independent prediction sets.

Their guarantees show that:
Under training poisoning, they show the conformal prediction set derived with smoothed score function is both size reliable and coverage reliable.
Furthermore, for calibration poisoning, they show under some conditions their majority prediction set method is coverage and size reliable.

Finally, they show the effectiveness of their approach experimentally, showing their method achieves reliability while maintaining utility on clean data.

**Strengths:**

I think overall, the idea of partitioning the training data and training different classifiers and aggregating their predictions is pretty natural and has been in around for both differential privacy and robustness against poisoning attacks. But anyways, they are the first to study it in the context of conformal prediction.

**Weaknesses:**

I think overall, the idea of partitioning the training data and training different classifiers and aggregating their predictions is pretty natural and has been in around for both differential privacy and robustness against poisoning attacks. But anyways, they are the first to study it in the context of conformal prediction.

**Questions:**

In equation 2, it might be good to remind the reader that K is the number of classes.

 I was confused by Definition 1, I think here you mean that the adversary flips y_i for some examples x_i, and then as a result, the prediction set C(x_{n+1}) might inflate or shrink. Is this correct?

---

> ### Author Response · Authors · 2024-11-19
> **Response to Reviewer 1V8P**
>
> Thank you for acknowledging the novelty of our approach for conformal prediction.  We would like to emphasize that existing methods for robust classification are not applicable for certifying conformal prediction due to two key differences: (1) prediction sets inherently involve multiple classes, and (2) they can be manipulated through poisoning during both training and calibration. We believe our contributions are technically solid and non-trivial, as also highlighted by other reviewers. Overall, we believe our work represents a significant step forward at the intersection of both communities (conformal prediction and robustness), providing a solid foundation for future research.
>
> **Regarding your questions.**  Regarding Definition 1, your understanding is correct. We design prediction sets reliable against adversaries that aim to inflate or shrink the prediction sets $C(x_{n+1})$ by changing the labeled datasets. As you suggested, we have also added a reminder that $K$ represents the number of classes in Equation 2.
>
> Thank you for your review! We hope that we could address all your questions to your satisfaction. Please let us know if you have any additional comments or questions.

---

> > ### Comment · Reviewer_1V8P · 2024-11-26
> >
> > I thank the authors for their response. At this point, I have no further questions or concerns, and I will consider increasing my score.

---

> > > ### Author Response · Authors · 2024-11-27
> > > **Response to Reviewer 1V8P**
> > >
> > > Thank you for confirming that there are no further questions or concerns. We would appreciate an updated score. Thank you again for your review.

---

### Author Response · Authors · 2024-11-19
**Global response**

We want to thank all reviewers for their valuable feedback. We changed our initial submission in response to their comments as follows:

- Reviewer AjMN: Few clarifications, and additional experiments in Appendix C.
- Reviewer HznX: Fixed typos, and additional experiments in Appendix B.
- Reviewer 1V8P: Additional reminder for the reader about parameter $K$ (number of classes).

---

### Meta-Review · Area_Chair_srwq · 2024-12-21

**Metareview:**

This work studies the robustness of conformal inference in cases of poisoning attacks in both training and calibration datasets, and devises a new method (dubbed Reliable Prediction Sets) that recovers analogous guarantees to conformal prediction in such cases. For corruption of training samples, the approach relies on methods that are well-known in the poisoning attacks community, such as splitting the data into disjoint subsets resulting in a "smooth" score function, which enjoys theoretical guarantees. For corruptions in calibration, the method is calibrated on different subsets of that data and aggregated via a majority vote. Their methodology is demonstrated on numerical experiments for image classification.

**Strengths**
* Novel problem formulation
* Important problem
* Novel approach
* Sound theoretical results

**Weaknesses**
None severe; all weaknesses noted by the reviewers were addressed by the authors in the rebuttal period.

**Summary** This is a good paper. The clarification on the impact of the method's parameters, related methods, as well as fixing some typos and small mistakes, have all improved the quality of the paper. I'm recommending Accept.

**Additional Comments On Reviewer Discussion:**

The exchange between authors and reviewers was productive. All of the weaknesses raised by the reviewers were minor, including small mistakes in definitions, typos, and clarification. All were addressed.

---

### Decision · Program_Chairs · 2025-01-22

Accept (Spotlight)